# Near-Optimal Quantum Algorithm for Minimizing the Maximal Loss

**Hao Wang**[*]
School of EECS
Peking University
Beijing, China
`tony.wanghao@stu.pku.edu.cn`

**Chenyi Zhang**[*]
Computer Science Department
Stanford University
Stanford, CA 94305, USA
`chenyiz@stanford.edu`

**Tongyang Li**
Center on Frontiers of Computing Studies
School of Computer Science
Peking University
Beijing, China
`tongyangli@pku.edu.cn`

## Abstract

The problem of minimizing the maximum of $N$ convex, Lipschitz functions plays significant roles in optimization and machine learning. It has a series of results, with the most recent one requiring $O(N\epsilon^{-2/3} + \epsilon^{-8/3})$ queries to a first-order oracle to compute an $\epsilon$-suboptimal point. On the other hand, quantum algorithms for optimization are rapidly advancing with speedups shown on many important optimization problems. In this paper, we conduct a systematic study for quantum algorithms and lower bounds for minimizing the maximum of $N$ convex, Lipschitz functions. On one hand, we develop quantum algorithms with an improved complexity bound of $\tilde{O}(\sqrt{N}\epsilon^{-5/3} + \epsilon^{-8/3})$.[1] On the other hand, we prove that quantum algorithms must take $\tilde{\Omega}(\sqrt{N}\epsilon^{-2/3})$ queries to a first order quantum oracle, showing that our dependence on $N$ is optimal up to poly-logarithmic factors.

## 1 Introduction

Consider the problem of minimizing the maximum of $N$ convex functions $f_1, \ldots, f_N$ where each $f_i \colon \mathbb{R}^d \to \mathbb{R}$ is convex and $L$-Lipschitz. Our goal is to find a point $x_\star$ satisfying

$$F_{\max}(x_\star) - \inf_{x \in \mathbb{R}^d} F_{\max}(x) \leq \epsilon \tag{1}$$

for some target accuracy $\epsilon$, where

$$F_{\max}(x) \coloneqq \max_{i \in [N]} f_i(x). \tag{2}$$

This problem has wide applications in machine learning and optimization. Specifically, it characterizes the problem of minimizing the maximum of loss functions in supervised learning. For instance, in support vector machines (SVMs), $f_i$'s are loss functions representing the negative margin of the $i^{\text{th}}$ data point (Vapnik, 1999; Clarkson et al., 2012; Hazan et al., 2011; Li et al., 2019). Furthermore, minimizing the maximum loss in classification provides advantageous effects on training speed and generalization (Shalev-Shwartz & Wexler, 2016). In optimization, it falls into the category of robust optimization (Beyer & Sendhoff, 2007; Ben-Tal et al., 2009) that studies optimization of the worst-case objective. As a concrete example, $F_{\max}(x) = \max_{p \in \Delta^N} \sum_{i \in [N]} p_i f_i(x)$ is an instance

---

[*]Equal contribution.
[1]Throughout this paper, $\tilde{O}$ omits poly-logarithmic factors, i.e., $\tilde{O}(f) = O(f \operatorname{poly}(\log f))$.

of distributionally robust optimization (Delage & Ye, 2010; Ben-Tal et al., 2013) with the uncertain set being the whole set $\Delta^N$ of probability distributions with cardinality $N$.

Given these motivations, Problem (1) has been extensively studied in previous literature. In particular, it is known that it can be solved by subgradient method using $O(\epsilon^{-2})$ iterations, e.g., see Nesterov (2018). In each iteration, however, $N$ queries are needed to compute the exact subgradient, which leads to an overall query complexity of order $O(N\epsilon^{-2})$. Furthermore, Carmon et al. (2021) proposed an algorithm based on the ball optimization acceleration scheme (Carmon et al., 2020b) that uses $\tilde{O}(N\epsilon^{-2/3} + \epsilon^{-8/3})$ subgradient queries. Carmon et al. (2021) also proved an $\Omega(N\epsilon^{-2/3})$ lower bound on the number of subgradient queries via the "chain construction" technique in optimization lower bounds (Carmon et al., 2020a; Diakonikolas & Guzmán, 2019; Guzmán & Nemirovski, 2015; Nemirovski & Yudin, 1983; Woodworth & Srebro, 2016), indicating that their algorithm is optimal to a poly-logarithmic factor in the parameter regime where $N \geq \epsilon^{-2}$.

On the other hand, quantum computing has been a rapidly advancing technology in recent years. The main motivation of studying quantum computing is its potential of solving problems significantly faster than the classical counterparts. The most famous example is Shor's algorithm Shor (1999) for factoring integers in polynomial time with high success probability on quantum computers. In optimization theory, various quantum algorithms for optimization problems have been developed, providing quantum speedups for semidefinite programs (Brandão & Svore, 2017; Brandão et al., 2019; van Apeldoorn & Gilyén, 2019; van Apeldoorn et al., 2020; Kerenidis & Prakash, 2020; Augustino et al., 2021), convex optimization (Chakrabarti et al., 2020; van Apeldoorn et al., 2020), nonconvex optimization (Liu et al., 2022; Gong et al., 2022; Childs et al., 2022; Zhang et al., 2021), etc. Conversely, quantum lower bounds on convex optimization (Garg et al., 2021; Garg et al., 2021) and nonconvex optimization (Zhang & Li, 2023) are also established. However, the problem of minimizing the maximum of functions is widely open in quantum computing at the moment.

**Contributions.** In this paper, we conduct a systematic study of quantum algorithms and lower bounds for minimizing the maximum of $N$ convex and Lipschitz functions $f_1, \ldots, f_N$. In particular, we assume the access to the following quantum zeroth-order oracle:

$$O_f \left| i \right\rangle \left| x \right\rangle \left| y \right\rangle \to \left| i \right\rangle \left| x \right\rangle \left| y + f_i(x) \right\rangle, \tag{3}$$

where $\left| \cdot \right\rangle$ denotes input or output register that allow queries in *quantum superpositions*, the essence of speedups from quantum algorithms. Specifically, a quantum algorithm can choose $x_1, \ldots, x_N \in \mathbb{R}^d$, $y_1, \ldots, y_N \in \mathbb{R}$, and $c \in \mathbb{C}^N$ such that $\sum_{i=1}^N |c_i|^2 = 1$, and applies the quantum oracle as follows:

$$O_f \left( \sum_{i=1}^N c_i \left| i \right\rangle \left| x_i \right\rangle \left| y_i \right\rangle \right) = \sum_{i=1}^N c_i \left| i \right\rangle \left| x_i \right\rangle \left| y_i + f_i(x) \right\rangle. \tag{4}$$

If we measure this quantum state, with probability $|\mathbf{c}_i|^2$ the third register gives $y_i + f_i(x)$. This oracle is a standard assumption by quantum algorithms for optimization, see e.g. Chakrabarti et al. (2020); van Apeldoorn et al. (2020); Liu et al. (2022); Gong et al. (2022); Zhang et al. (2021). More introductions to notations and definitions of quantum computing are given in Section 2.

**Theorem 1** (Main theorem). *There is a quantum algorithm (Algorithm 1) that outputs an $x_\star$ satisfying Eq. (1) with probability at least $2/3$ using $\tilde{O}(\sqrt{N}\epsilon^{-5/3} + \epsilon^{-8/3})$ queries to the $O_f$ in Eq. (3). On the other hand, such quantum algorithms must take $\tilde{\Omega}(\sqrt{N}\epsilon^{-2/3})$ queries to $O_f$ (Theorem 4).*

Compared to the state-of-the-art classical algorithm for minimizing the maximum of $N$ convex and Lipschitz functions, we achieve quadratic quantum speedup in the number of functions $N$. On the other hand, our quantum lower bound $\Omega(\sqrt{N}\epsilon^{-2/3})$ establishes the near-optimality of our quantum algorithm in $N$. See Table 1 for more detailed comparisons.

**Techniques.** Our quantum algorithm follows the scheme of Carmon et al. (2021), while we achieve quantum speedup by leveraging quantum samples. As a starting point, classical algorithms (Carmon et al., 2021; Nesterov, 2005) proceed by considering the so-called "softmax" approximation of the maximum, defined as

$$F_{\text{smax},\epsilon}(x) := \epsilon' \log \left( \sum_{i \in [N]} \exp \left( \frac{f_i(x)}{\epsilon'} \right) \right), \qquad \epsilon' = \frac{\epsilon}{2 \log N}. \tag{5}$$

Table 1: Summary of results on minimizing the maximum of $N$ convex, Lipschitz functions to accuracy $\epsilon$.

| Reference | Method | Oracle | Upper bound | Lower bound |
|---|---|---|---|---|
| Nesterov (2018) | Subgradient method | Classical First-Order | $O(N\epsilon^{-2})$ | — |
| Carmon et al. (2021) | Ball optimization acceleration | Classical First-Order | $O(N\epsilon^{-2/3} + \epsilon^{-8/3})$ | $\Omega(N\epsilon^{-2/3} + \epsilon^{-2})$ |
| **this work** | Ball optimization acceleration with quantum sampling | Quantum Zeroth-Order | $\tilde{O}(\sqrt{N}\epsilon^{-5/3} + \epsilon^{-8/3})$ | $\Omega(\sqrt{N}\epsilon^{-2/3})$ |

Note that the choice of $\epsilon'$ here promises that a solution $x_\star$ of

$$F_{\text{smax},\epsilon}(x_\star) - \inf_{x\in\mathbb{R}^d} F_{\text{smax},\epsilon}(x) \leq \frac{\epsilon}{2} \tag{6}$$

will automatically satisfy (1) (see Lemma 4). Our algorithm has two stages:

- Implement a regularized ball optimization oracle (BROO) of $F_{\text{smax},\epsilon}$ that when query at $\mathbf{x}$ returns an (approximate) minimizer of $F_{\text{smax},\epsilon}$ in a ball of radius $r$ around $\mathbf{x}$; and then
- Find an approximate minimum of $F_{\text{smax},\epsilon}$ using ball optimization oracle via a ball optimization algorithm (Carmon et al., 2020b).

Here, the ball optimization algorithm in Carmon et al. (2020b) is a variant of the Monteiro-Svaiter acceleration (Bubeck et al., 2019; Bullins, 2020; Gasnikov et al., 2019; Monteiro & Svaiter, 2013) and can minimize $f$ to accuracy $\epsilon$ using $\tilde{O}\left(r^{-2/3}\right)$ queries to the BROO. Carmon et al. (2021) showed that BROO for $F_{\text{smax},\epsilon}$ can be implemented by considering instead a "softened" function

$$\Gamma_\epsilon(x) = \sum_{i\in[N]} p_i \epsilon' \cdot e^{\frac{f_i(x) - f_i(\bar{x})}{\epsilon'}}, \qquad p_i = \frac{e^{f_i(\bar{x})/\epsilon'}}{\sum_{j\in[N]} e^{f_i(\bar{x})/\epsilon'}}, \tag{7}$$

where $\epsilon' = \epsilon/(2\log N)$, which shares the same minimizer as $F_{\text{smax},\epsilon}$ and has a finite sum structure enables stochastic gradient methods. In their implementation, the main bottleneck for $N$ arises from the sampling step of the distribution $p_i$. To obtain $T$ samples, the algorithm requires precomputing all $p_1,\ldots,p_N$, which takes $\Omega(N)$ classical queries and cannot be further accelerated in the classical setting. However, this task shares a similar formula with the *Gibbs sampling problem*, which exhibits quantum speedups (Bouland et al., 2023; Gao et al., 2023). Building on this intuition, we demonstrate that BROO can be executed using just $\tilde{O}(\sqrt{N})$ queries to the quantum oracle $O_f$ defined in (3). Our quantum algorithm begins by preparing $T$ copies of the quantum state $\sum_i \sqrt{p_i} \,|i\rangle$. The first step involves identifying the $K$ largest $p_i$ values and corresponding indices, which can be accomplished with only $\sqrt{NT}$ queries to the quantum oracle $O_f$, in contrast to the classical case's query complexity of $\Omega(N)$. This quadratic speedup in $N$ contributes to our overall quantum speedup.

From a high-level perspective, quantum speedups in optimization algorithms have been mainly investigated in gradient descent methods (Liu et al., 2022; Gong et al., 2022; Zhang et al., 2021), cutting plane methods (Chakrabarti et al., 2020; van Apeldoorn et al., 2020), interior point methods (Kerenidis & Prakash, 2020; Augustino et al., 2021), etc., but quantum algorithms based on trust region methods are widely open. Our result can be seen as a first attempt for quantum algorithms using trust region methods with speedup.

We establish our quantum lower bound for minimizing the maximal loss by applying a quantum progress control method to the classical hard instance described in Carmon et al. (2021). The quantum progress control method, initially introduced in Garg et al. (2021), has been widely used in proving quantum lower bounds for optimization problems, including non-smooth convex optimization (Garg et al., 2021), smooth convex optimization (Garg et al., 2021), and (stochastic) non-convex optimization (Zhang & Li, 2023). This method is analogous to the classical progress control technique employed in proving classical lower bounds for optimization problems (Carmon et al., 2020a; Arjevani et al., 2020; 2022; Bubeck et al., 2019), and is based on demonstrating that any algorithm attempting to solve the optimization problem must acquire a sufficient amount of information through an adaptive "chain structure". This chain structure is a well-established concept in classical optimization lower bound proofs, as seen in prior works such as (Carmon et al., 2020a; Nemirovski & Yudin, 1983; Woodworth & Srebro, 2016; Guzmán & Nemirovski, 2015; Diakonikolas & Guzmán, 2019).

The quantum progress control method proceeds by modeling any quantum algorithm as a sequence of unitaries represented as

$$\cdots V_3 O_f V_2 O_f V_1 O_f V_0$$

applied to the initial state $|0\rangle$, where $O_f$ is the oracle encoding the information of $f$ and $V_i$'s are unitaries that are independent from $f$. Then, one can show that those queries can only learn the information and make progress along the chain structure in an *adaptive* manner. This ultimately leads to a lower bound that scales in proportion to the length of the chain multiplied by the cost to make a unit progress along the chain.

In the majority of cases, the query cost of making a unit progress along the chain is just 1, see e.g., Garg et al. (2021); Garg et al. (2021). Nevertheless, there exists methods to incorporate more "hardness" into the chain by constructing hard instances where making progress along the chain requires solving a subproblem. For example, the quantum stochastic lower bound for nonconvex optimization algorithm (Zhang & Li, 2023) proceeds by applying a hard instance with a "chain structure" such that, at each point where any algorithm attempts to progress by one step of unit length along the chain, it is required to solve a multivariate mean estimation problem, whose quantum lower bound is given in Cornelissen et al. (2022).

Our lower bound builds upon the same framework. In particular, we demonstrate that, at each point where a quantum algorithm attempts to progress by one step along the chain of the hard instance defined in described in Carmon et al. (2021), it has to solve an unstructured search problem. This intuition, however, cannot work straightforwardly. This arises from the fact that even in the case of a completely random guess, any algorithm possesses a success probability that is at least polynomially small for solving the unstructured search problem and progress by multiple steps along the chain. This success probability, although small, poses a challenge to the progress control argument where the probability of making progress by multiple steps at once should be super-polynomially small.

We address this issue by introducing a multi-round version of unstructured search problem. In this modified problem, each round's solution acts as a "key" that unlocks the next round, forcing it to solve these unstructured search problems adaptively. We demonstrate that for any quantum algorithm making a maximum of $O(N)$ queries to this multi-round unstructured search problem, its overall success probability in solving the entire multi-round problem, i.e., solve the unstructured search problem of each round, is only super-polynomially small. After establishing the lower bound for the multi-round unstructured search problem, we show that making progress along the chain for a given length is as hard as solving the multi-round unstructured search problem of the same number of rounds. Formally, we show that any quantum algorithm has to use $\tilde{\Omega}(\sqrt{N})$ queries to make $O(\text{poly}\log(1/\epsilon))$ progress along chain of length $\Omega\left(\epsilon^{-2/3}\right)$, giving our $\tilde{\Omega}\left(\sqrt{N}\epsilon^{-2/3}\right)$ lower bound.

**Open questions.** Our paper leaves several natural open questions for future investigation:

- Can we narrow or even close the gap of order $\epsilon$ between the leading complexity term in our quantum algorithm and our quantum lower bound? Useful techniques might be found in quantum algorithms for Gibbs sampling (Bouland et al., 2023; Gao et al., 2023), but it remains uncertain if we can reconcile the differences in the two settings.

- It is shown in Carmon et al. (2021) that the smoothness of $f_i$ can enable faster classical algorithms for minimizing the maximum loss, even in the case where the smoothness parameter is of order $1/\epsilon$ and all the $f_i$ are almost non-smooth. The intuition is that we can apply the accelerated variance reduction method (Allen-Zhu, 2017) to implement the ball optimization oracle of $F_{\text{smax},\epsilon}$ if it is smooth, which contains a mean estimation subroutine that cannot be directly improved by quantum algorithms (Cornelissen et al., 2022; Zhang & Li, 2023). Hence, a natural question to ask is if there exist quantum algorithms that can utilize the smoothness structure and provide better convergence rates.

## 2 PRELIMINARIES

**Basic notations in quantum computing.** Quantum mechanics can be formulated in the language of linear algebra. Specifically, we define the computational basis of the space $\mathbb{C}^d$ by $\{\vec{e}_1, \ldots, \vec{e}_d\}$,

where $\vec{e}_i = (0, \ldots, 1, \ldots, 0)^\top$ with the $i^{\text{th}}$ coordinate being 1 and other coordinates being 0. These basis vectors can be written by the *Dirac notation*, where we write $\vec{e}_i$ as $|i\rangle$, and write $\vec{e}_i^\top$ as $\langle i|$.

A *d-dimensional quantum state* $|v\rangle = (v_1, \ldots, v_d)^\top$ is a unit vector in $\mathbb{C}^d$, i.e., $\sum_{i=0}^{d-1} |v_i|^2 = 1$. For each $i$, $v_i$ is named the *amplitude* in $|i\rangle$. If at least two amplitudes are nonzero, we say $|v\rangle$ is in *superposition* of the computational basis.

*Tensor product* of quantum states is their Kronecker product: if $|u\rangle \in \mathbb{C}^{d_1}$ and $|v\rangle \in \mathbb{C}^{d_2}$, then $|u\rangle \otimes |v\rangle = (u_1 v_1, u_1 v_2, \ldots, u_{d_1} v_{d_2})^\top \in \mathbb{C}^{d_1} \otimes \mathbb{C}^{d_2}$. $|u\rangle \otimes |v\rangle$ can be abbreviated as $|u\rangle |v\rangle$.

The basic element in classical computing is one bit. Following this pattern, the basic element in quantum computing is one *qubit*, which is a quantum state in $\mathbb{C}^2$. Formally, a qubit state has the form $a|0\rangle + b|1\rangle$ where $a, b \in \mathbb{C}, |a|^2 + |b|^2 = 1$. Furthermore, an *n-qubit state* has the form $|v_1\rangle \otimes \cdots \otimes |v_n\rangle$, where each $|v_i\rangle$ is a qubit state for all $i \in [n]$. As a result, $n$-qubit states form a Hilbert space of dimension $2^n$.

Calculations in quantum computing are *unitary transformations* and can be stated in the circuit model where a *k-qubit gate* is a unitary matrix in $\mathbb{C}^{2^k}$. Two-qubit quantum gates are *universal*, i.e., every $n$-qubit gate can be written as the product of a series of two-qubit gates. Therefore, the *gate complexity* of a quantum algorithm can be regarded as its number of two-qubit gates.

**Basic quantum algorithms.** In our paper, we use the following quantum algorithms in previous works as basic building blocks of our quantum algorithm.

**Proposition 1** (Top-K Maximum Finding - Dürr et al. 2006, Theorem 4.2). *Let* $K, N \in \mathbb{N}$, $1 \leq K \leq N$, $\delta \in (0, 1)$ *and* $\mathbf{w} \in \mathbb{R}_{\geq 0}^N$. *There exists a quantum algorithm* QuantumMaximumFinding$(K, N, \mathbf{w}, \delta)$ *that outputs the positions of $K$ largest entries in $\mathbf{w}$ with success probability at least $1 - \delta$. The algorithm performs $O(\sqrt{KN} \log(1/\delta))$ queries the following quantum oracle $O_\mathbf{w}$ that encodes $\mathbf{w}$:*

$$O_\mathbf{w} |i\rangle |y\rangle \to |i\rangle |y + w_i\rangle. \tag{8}$$

**Proposition 2** (State Preparation - Zalka 1998; Grover & Rudolph 2002; Kaye & Mosca 2001). *Given integers $N, T$, a non-zero vector $\mathbf{w} \in \mathbb{R}_{\geq 0}^N$ with $\|\mathbf{w}\|_1 = 1$ and a classical procedure that computes $\sum_{l=i}^{j} w_j, \forall i \leq j$, there exists a quantum procedure* QuantumStatePreparation$(N, T, \mathbf{w})$ *that outputs the classical description of a quantum circuit $\mathcal{D}$ that satisfies $\mathcal{D} |0\rangle \to |\mathbf{w}\rangle = \sum_i \sqrt{w_i} |i\rangle$.*

**Proposition 3** (Amplitude Amplification - Brassard et al. 2002, Theorem 3). *Let $\mathcal{C}$ be a quantum circuit that prepares the state $\mathcal{C} |0\rangle = \sqrt{p} |\phi\rangle |0\rangle + \sqrt{1-p} |\phi_\perp\rangle |1\rangle$ for some $p \in [0, 1]$ and two unit states $|\phi\rangle, |\phi_\perp\rangle$. There exists a quantum algorithm* AmplitudeAmplification$(\mathcal{C})$ *that outputs the state $|\phi\rangle$ by using $O(1/\sqrt{p})$ calls of $\mathcal{C}$ and $\mathcal{C}^\dagger$ in expectation.*

**Proposition 4** (Quantum Gradient Estimation - Jordan 2005, Lemma 2.2). *Given access to the quantum zeroth-order oracle $O_f$ defined in (3), there exists a quantum algorithm* QuantumGradientEstimation$(i, x)$ *that outputs the gradient $\nabla f_i(x)$ using only one query to $O_f$.*

## 3  QUANTUM ALGORITHM FOR MINIMIZING THE MAXIMAL LOSS

In this section, we present our quantum algorithm for minimizing the maximal loss. Our approach builds upon the framework of (Carmon et al., 2021, Algorithm 1), which proceeds by preparing a *ball optimization oracle* that can minimize the objective function in a very small region, and then optimize the objective function by using this oracle recursively. In particular, (Carmon et al., 2021, Algorithm 1) is a variant of Carmon et al. (2020b) and utilizes the Monteiro-Svaiter acceleration (Monteiro & Svaiter, 2013). Notably, there has been a series of work (Carmon et al., 2020b; 2021) on designing optimization algorithms following this framework, given that it is natural in some problems where we can access the ball optimization oracle efficiently. Compared to (Carmon et al., 2021, Algorithm 1), our algorithm differs by a critical observation that the bottleneck in the classic algorithm, mainly the sampling, can be sped up using a quantum subroutine based on Gibbs sampling.

As in Carmon et al. (2021), we consider the ball regularized optimization oracle (BROO), which is a generalization from the stricter ball optimization oracle Carmon et al. (2020b). Formally, a BROO is defined as follows.

**Definition 1** (Carmon et al. 2021, Definition 1). *We say that a mapping $O_{\lambda,\delta}$ is a Ball Regularized Optimization Oracle of radius $r$ ($r$-BROO) for $f$, if for every query point $\bar{x}$, regularization parameter $\lambda$ and desired accuracy $\delta$, it returns $\tilde{x} = O_{\lambda,\delta}(\bar{x})$ satisfying*

$$f(\tilde{x}) + \frac{\lambda}{2}\|\tilde{x} - \bar{x}\|^2 \le \min_{x \in \mathbb{B}_r(\bar{x})} \left\{ f(x) + \frac{\lambda}{2}\|x - \bar{x}\|^2 \right\} + \frac{\lambda}{2}\delta^2. \tag{9}$$

The following result illustrates the convergence rate achieved by the classical ball optimization algorithm (Carmon et al., 2021, Algorithm 1) using BROO queries.

**Proposition 5** (BROO Acceleration - Rephrased from Carmon et al. 2021, Theorem 1). *Let $f : \mathbb{R}^d \to \mathbb{R}$ be convex and $L_f$-Lipschitz, and let $z \in \mathbb{R}^d$. For any domain bound $R > 0$ accuracy level $\epsilon > 0$, and initial point $x_0 \in \mathbb{R}^d$, there exists a (classical) algorithm that returns a point $x \in \mathbb{R}^d$ satisfying $f(x) - \min_{z \in B_R(x_0)} f(z) \le \epsilon/2$ using at most*

$$O\left( \left(\frac{R}{r_\epsilon}\right)^{2/3} \log^2\left(\frac{R}{r_\epsilon}\right) \right) \tag{10}$$

*queries to an $r_\epsilon$-BROO, in which $r_\epsilon = \epsilon/2L_f \log N$. The algorithm also guarantees that the BROO query parameters $(\lambda, \delta)$ satisfy $\Omega\left(\frac{\epsilon}{r_\epsilon R}\right) \le \lambda \le O\left(\frac{L_f}{r_\epsilon}\right)$ and $\delta = \Omega\left(\frac{\epsilon}{\lambda R}\right)$.*

### 3.1 Quantum BROO implementation of $F_{\mathrm{smax},\epsilon}$

In this subsection, we present a quantum algorithm (Algorithm 1) that implements a BROO given access to the oracle $O_f$ defined in (3). The query complexity of Algorithm 1 is given in Theorem 2.

---

**Algorithm 1:** Quantum-Epoch-SGD-Proj on the exponentiated softmax

**Input:** Functions $f_1, \ldots, f_N$, ball center $\bar{x}$, ball radius $r_\epsilon$, regularization strength $\lambda$, smoothing parameter $\epsilon'$, failure probability $\sigma$

**Parameters:** Step size $\eta_1 = 1/(3\lambda)$, domain size $D_1 = \Theta(G\sqrt{\log(\log(T)/\sigma)}/\lambda)$, $T_1 = 450$ and total iteration budget $T$

**Output:** Approximate minimizer of $\Gamma_{\epsilon,\lambda}$ (and hence $F_{\mathrm{smax},\epsilon}^\lambda$ in $\mathbb{B}_{r_\epsilon}(\bar{x})$)

1 Prepare $T$ identical states $|\psi\rangle = \sum_i e^{f_i(\bar{x})/2\epsilon'} / \sqrt{\sum_{i\in[N]} e^{f_i(\bar{x})/\epsilon'}} |i\rangle$ using Algorithm 2
2 Initialize $x_1^1 \in \mathbb{B}_{r_\epsilon}(\bar{x})$ arbitrarily, set $k = 1$
3 **while** $\sum_{i\in[k]} T_i \le T$ **do**
4    **for** $t = 1, \ldots, T_k$ **do**
5       Sample $i \in [N]$ by measuring the next unmeasured $|\psi\rangle$
6       Call `QuantumGradientEstimation`$(i, x)$ to compute
         $\nabla f_i^\lambda(x) = \nabla f_i(x) + \lambda(x - \bar{x})$
7       $\hat{g}_t = e^{(f_i^\lambda(x) - f_i^\lambda(\bar{x}))/\epsilon'} \nabla f_i^\lambda(x)$
8       Update $x_{t+1}^k \leftarrow \Pi_{B_r(\bar{x}) \cap B_{D_k}(x_1^k)}(x_t^k - \eta_k \hat{g}_t)$
9    Let $x_1^{k+1} \leftarrow \frac{1}{T_k} \sum_{t\in[T_k]} x_t^k$
10    Update parameters $T_{k+1} \leftarrow 2T_k$, $\eta_{k+1} \leftarrow \eta_k/2$, $D_{k+1} \leftarrow D_k/\sqrt{2}$, $k \leftarrow k + 1$
11 **return** $x_1^k$

---

**Theorem 2.** *Let $f_1, f_2, \ldots, f_N$ be $L_f$-Lipschitz functions. Let $\sigma \in (0,1)$, $\epsilon, \delta > 0$ and $r_\epsilon = \epsilon/2L_f \log N$. For any $\bar{x} \in \mathbb{R}^d$ and $\lambda \le O(L_f/r_\epsilon)$, with probability at least $1 - 2\sigma$, Algorithm 1 outputs a valid $r_\epsilon$-BROO response for $F_{\mathrm{smax},\epsilon}$ to query $\bar{x}$ with regularization $\lambda$ and accuracy $\delta$, and has cost*

$$O(\sqrt{N\mathcal{T}} \log(1/\sigma) + \mathcal{T}) \tag{11}$$

*in which $\mathcal{T} = L_f^2 \lambda^{-2} \delta^{-2} \log(\log(L_f/\lambda\delta)/\sigma)$.*

Note that when implementing BROO we are not essentially minimizing $F_{\mathrm{smax},\epsilon}(x)$ but instead

$$F_{\mathrm{smax},\epsilon}^\lambda(x) := F_{\mathrm{smax},\epsilon}(x) + \lambda/2 \cdot \|x - \bar{x}\|^2 = \epsilon' \log\left( \sum_{i\in[N]} \exp\left(f_i^\lambda(x)/\epsilon'\right) \right), \tag{12}$$

in which $f_i^\lambda(x) := f_i(x) + \lambda/2 \cdot \|x - \bar{x}\|^2$ is a regularized version of $f_i(x)$. We define its "softened" function $\Gamma_{\epsilon,\lambda}$ (Eq. (7)) accordingly.

Classically, a BROO of $F_{\mathrm{smax},\epsilon}$ can be implemented by (Carmon et al., 2021, Algorithm 2) using Epoch-SGD-Proj proposed in Hazan & Kale (2014) with the stochastic gradient oracle implemented by sampling an $i$ first and evaluate the gradient of one of the $\gamma_i$'s. This process is similar to stochastic gradient descent but introduces epochs, where a intra-epoch mirror descent takes place. Additionally, a projection is placed at the end of each iteration such that the variance of the stochastic gradient can be limited and probability bounds can be obtained rather than expectation bounds.

In comparison, our quantum algorithm (Algorithm 1) is based on the same framework, with a critical change. Rather than sampling classically from the same distribution many times within each epoch, we derive a quantum subroutine (Algorithm 2) that sped up this bottleneck exploiting quantum advantage on Gibbs sampling. Additionally, comparing to the classic algorithm, which requires first-order oracle to acquire gradient information, the proposed algorithm uses `QuantumGradientEstimation` (Proposition 4) to efficiently estimate the gradient through quantum zeroth-order oracle.

---

**Algorithm 2:** Quantum sampling of the softmax distribution.

**Input:** Number of copies $K$, failure probability $\delta$

1  Denote $\mathbf{w} = (f_1(\bar{x}), \ldots, f_N(\bar{x}))^\top$. Call `QuantumMaximumFinding`$(K, N, \mathbf{w}, \delta)$ to compute the set $H \subseteq [N]$ of the positions of $K$ largest entries in $f_i(\bar{x})$ with failure probability $\delta$.

2  Compute $h = \min_{i \in H} f_i(\bar{x})$ and

$$\mathcal{Z} = (N - K)\exp(h/\epsilon') + \sum_{i \in H}\exp(f_i(\bar{x})/\epsilon')$$

3  Denote $\mathbf{w}' = (w'_1, \ldots, w'_N)$ where

$$w'_i = \begin{cases} \frac{\exp(f_i(\bar{x})/\epsilon')}{\mathcal{Z}}, & i \in H \\ \frac{\exp(h/\epsilon')}{\mathcal{Z}}, & i \notin H, \end{cases} \qquad \forall i \in [N],$$

call `QuantumStatePreparation`$(N, T, \mathbf{w}')$ to build a unitary $\mathcal{D}$ such that

$$|0\rangle \xrightarrow{\mathcal{D}} \sum_{i \in H}\sqrt{\frac{\exp(f_i(\bar{x})/\epsilon')}{\mathcal{Z}}}|i\rangle + \sum_{i \notin H}\sqrt{\frac{\exp(h/\epsilon')}{\mathcal{Z}}}|i\rangle$$

4  Construct the circuit $\mathcal{C}$, which first applies $\mathcal{D}$ to the output register, as represented in Figure 1.
5  Call `AmplitudeAmplification`$(\mathcal{C})$ $K$ times and return the $K$ output states.

---

In Line 4, the circuit $\mathcal{C}$ is constructed from the unitary $\mathcal{D}$ in Line 3. Amplitude amplification is then applied to prepare the state we need by $\mathcal{C}$. More explanation of Algorithm 2 is given in Appendix A. The lemma below shows quantum speedup for producing multiple samples from the same distribution.

**Lemma 1.** *With probability at least $1 - \delta$, Algorithm 2 on input $K, \delta$ produces $K$ samples from the probability distribution*

$$p_i = \frac{\exp(f_i(\bar{x})/\epsilon')}{\sum_{j \in [N]}\exp(f_j(\bar{x})/\epsilon')}, \qquad i = 1, 2, \ldots, N, \tag{13}$$

*using $O(\sqrt{NK}\log(1/\delta))$ queries to the quantum evaluation oracle $U_f$ defined in (3).*

We present the proof of this lemma in Appendix A. Next, we cite the result regarding the convergence rate of stochastic gradient method with projection for $\mu$-strongly convex functions.

**Lemma 2** (Hazan & Kale 2014, Theorem 11)**.** *Let $f : X \to \mathbb{R}$ be a $\mu$-strongly-convex objective function on a convex compact set $X$ with minimizer $x_\star$. With an unbiased stochastic estimator with norm bounded by $G$, Epoch-SGD-Proj algorithm uses $T$ stochastic gradient queries and finds an approximate minimizer $\tilde{x}$ satisfying with probability $1 - \sigma$*

$$f(\tilde{x}) - f(x_\star) \le O\left(\frac{G^2\log(\log(T)/\sigma)}{\mu T}\right). \tag{14}$$

*Proof of Theorem 2.* According to (7), the gradient of $\Gamma_{\epsilon,\lambda}(x)$ can be expressed as follows:

$$\nabla\Gamma_{\epsilon,\lambda}(x) = \sum_i p_i e^{(f_i^\lambda(x) - f_i^\lambda(\bar{x}))/\epsilon'}\nabla f_i(x). \tag{15}$$

Thus, we have $\mathbb{E}[\hat{g}_t] = \nabla\Gamma_{\epsilon,\lambda}(x_t)$. Moreover, by Lemma 5, the quantity $e^{(f_i^\lambda(x) - f_i^\lambda(\bar{x}))/\epsilon'}\nabla f_i(x)$ is bounded by a constant $G = O(L_f)$. Thus, $\hat{g}_t$ is a valid stochastic gradient.

Next, by applying Lemma 2 with $T = \Theta(L_f^2\lambda^{-2}\delta^{-2}\log(\log(L_f/\lambda\delta)/\sigma))$, Algorithm 1 outputs a minimizing point $x_\star$ of $\Gamma_{\epsilon,\lambda}$ with suboptimality

$$\Gamma_{\epsilon,\lambda}(x_\star) - \min_{x\in\mathbb{R}^d}\Gamma_{\epsilon,\lambda}(x) \le O\left(\frac{L_f^2}{\lambda T}\log(\log T/\sigma)\right) \le \frac{\lambda\delta}{6e^2}. \tag{16}$$

With Lemma 5, we can bound the suboptimality of $x_\star$ on $F_{\text{smax},\epsilon}^\lambda$ by $3e^2\lambda\delta/6e^2 \le \lambda\delta/2$, indicating that Algorithm 1 is a valid BROO. The probability of Algorithm 1 failing is at most $2\sigma$ considering either the Epoch-SGD-Proj fails (Lemma 2) or the quantum sampling fails (Lemma 1). □

## 3.2 Quantum Query Complexity of Minimizing the Maximal Loss

With our BROO oracle implementation at hand, we present our main result in this section, which describes the quantum query complexity of minimizing the maximum loss.

**Theorem 3.** *Let $f_1, f_2, \ldots, f_N$ be $L_f$-Lipschitz, let $x_\star$ be a minimizer of $F_{\max}(x) = \max_{i\in[N]}f_i(x)$ and assume $\|x_0 - x_\star\| \le R$ for a given initial point $x_0$ and some $R > 0$. For any $\epsilon > 0$, using the algorithm in Proposition 5 the BROO implementation for $F_{\text{smax},\epsilon}$ in Algorithm 1 finds a point with suboptimality $\epsilon$ with probability at least $\frac{2}{3}$ and has computational cost*

$$O(K^{5/3}\log^3 K(\sqrt{N\log K} + L_f R/\epsilon)/\log N) \tag{17}$$

*in which $K = L_f R\log(N)/\epsilon$.*

*Proof.* We use Proposition 5 along with Theorem 2 on the minimization problem to prove both the correctness and the overall query complexity of the algorithm. In particular, using the guarantees in Proposition 5, we can see that the results of Theorem 2 applies directly since $\lambda \le O(L_f/r_\epsilon)$. Letting $T$ be the bound of the total number of oracle queries in Proposition 5. We could have $\sigma = 1/6T$ such that the probability of Algorithm 1 not failing throughout all the queries is at least $2/3$. Thus, according to Proposition 5 the algorithm can output a minimizing point $x_\star$ for $F_{\text{smax},\epsilon}$ with suboptimality at most $\epsilon/2$. Thus, using Lemma 4, we conclude that $x_\star$ is a solution to (1), proving the correctness of the algorithm.

As for the query complexity, notice that from Proposition 5 the total number of calls to the BROO is

$$T = O\left(\left(\frac{L_f R\log N}{\epsilon}\right)^{2/3}\log^2\left(\frac{L_f R\log N}{\epsilon}\right)\right). \tag{18}$$

Using our implementation of the BROO (Algorithm 1), the total number of calls to the quantum oracle $U_f$ per BROO call is

$$O\left(\sqrt{N\log\left(\frac{L_f R\log N}{\epsilon}\right)}\left(\frac{L_f R}{\epsilon}\right)\log\left(\frac{L_f R\log N}{\epsilon}\right) + \left(\frac{L_f R}{\epsilon}\right)^2\log\left(\frac{L_f R\log N}{\epsilon}\right)\right) \tag{19}$$

in which we substitute $\delta = \Omega(\epsilon/\lambda R)$ and $\sigma = 1/6T$. □

## 4 Quantum Lower Bound for Minimizing the Maximum Loss

In this section, we establish a quantum query lower bound to demonstrate the optimality of our algorithm in Section 3 with respect to $N$. Classically, the query lower bound for minimizing maximal loss is established by a progress control argument given in Carmon et al. (2021). For any $x \in \mathbb{R}^d$, they define a quantity *progress* as

$$\text{prog}_\alpha(x) := \max\{i \ge 1 \,|\, |x_i| \ge \alpha\}. \tag{20}$$

Intuitively, the progress is the highest coordinate index that an algorithm discovers when it reaches $x$. Following that, Carmon et al. (2021) extended the concept of robust zero chain introduced in Carmon et al. (2020a) to the setting of minimization of maximum loss.

**Definition 2** (Carmon et al. 2021, Definition 2). *A sequence $f_1, \ldots, f_N$ of functions $f_i \colon \mathbb{B}_R(0) \to \mathbb{R}$ is called an $\alpha$-robust $N$-element zero-chain if for all $x \in \mathbb{B}_R(0)$, all $y$ in a neighborhood of $x$, and all $i \in [N]$, we have*

$$\mathrm{prog}_\alpha(x) \le p \implies f_i(y) = \begin{cases} f_i(y_{[\le p]}) & i < p+1 \\ f_i(y_{[\le p+1]}) & i = p+1 \\ f_N(y_{[\le p]}) & i > p+1, \end{cases} \tag{21}$$

*where for any $y \in \mathbb{R}^d$, $y_{[\le l]}$ denotes the vector whose first $l$ coordinates are identical to those of $y$ and the remained coordinates are zero.*

Carmon et al. (2021) demonstrates that, for a sequence of functions $f_1, \ldots, f_N$ forming an $\alpha$-robust $N$-element zero-chain with randomly permuted indices, a classical algorithm requires a lower bound of $\Omega(N)$ queries to increase the progress by 1 in expectation. Similarly, we show that it takes $\Omega(\sqrt{N})$ queries for a quantum algorithm to make progress on an $\alpha$-robust $N$-element zero-chain. Notably, the progress achieved after $\Omega(\sqrt{N})$ quantum queries exhibits a sub-exponential tail, and the probability of achieving super-logarithmic progress is super-polynomially small.

In this section, we first present in Section 4.1 a quantum analog of the progress control argument in Carmon et al. (2021). Finally, we introduce the main lower bound statement in Section 4.2.

## 4.1 PROGRESS CONTROL ARGUMENT FOR QUANTUM ALGORITHMS

We establish a quantum analog of the progress control argument given in Carmon et al. (2021). Following the notion of Garg et al. (2021); Garg et al. (2021); Zhang & Li (2023), we represent any quantum algorithm making $k$ queries to the oracle $O_f$ in the form of sequences of unitaries

$$A_{\mathrm{quan}}[k] = V_k O_f V_{k-1} O_f \cdots O_f V_1 O_f V_0 \tag{22}$$

applied to the initial state $|0\rangle$, followed by a measurement. Similar to Carmon et al. (2021, Proposition 1), we prove the following result that no quantum algorithm can guess the final column of $U$ or make progress efficiently (proof deferred to Appendix B.2).

**Proposition 6.** *Let $\delta, \alpha \in (0, 1)$ and let $N, T \in \mathbb{N}$ with $T \le N/2$. Let $\{f_i\}_{i \in [N]}$ be an $\alpha$-robust $N$-element zero-chain with domain $\mathbb{B}_R(0)$. For $d \ge T + \max(32T^3 \log(32\sqrt{N}T^5), 32T^3 \log(4T/\delta))$, draw $U$ uniformly from the set of $d \times T$ orthogonal matrices, and draw $\Pi$ uniformly from the set of permutations of $[N]$. Let $\tilde{f}_i(x) \coloneqq f_{\Pi(i)}(U^\top x)$. For any $t$-query quantum algorithm $A_{\mathrm{quan}}[t]$ equipped with an oracle on $\tilde{f}_1(x), \ldots, \tilde{f}_N(x)$, denote $x_t$ as its output. Then with probability at least $1 - \delta$ we have*

$$\mathrm{prog}_\alpha(U^\top x_t) < T, \qquad \forall t \le \frac{T\sqrt{N}}{CK^2}$$

*in which $C$ is a constant and $K = 4 \log T + \frac{1}{2} \log N + 4$.*

## 4.2 THE LOWER BOUND STATEMENT

Finally, we present the main theorem of the lower bound argument.

**Theorem 4.** *For any $L_f, L_g, R > 0$, $\epsilon < \min(L_f R, L_g R^2)$, $N \in \mathbb{N}$ and $\delta \in (0, 1)$, there exists a constant $C$ such that, for any quantum algorithm $A$ making less than $C \cdot \sqrt{N} \epsilon^{-2/3} \log N/\epsilon$ queries to the oracle $O_f$ defined in (3) and outputs a point $x_{out}$, there exists $L_f$-Lipschitz and $L_g$-smooth functions $(f_i)_{i \in [N]}$ with domain $B_R^d(0)$ for $d \ge T + \max(32T^3 \log(32\sqrt{N}T^5), 32T^3 \log(4T/\delta))$ such that*

$$\Pr\left[ F_{\max}(x_{out}) - \min_{x \in B_R^d(0)} F_{\max}(x) \ge \epsilon \right] \le \frac{1}{3}. \tag{23}$$

Theorem 4 implies that the problem can only be solved by any quantum algorithm using $\Omega(\sqrt{N} \epsilon^{-2/3})$ queries. The proof is deferred to Appendix B.3, but the idea of the proof is straightforward by combining Proposition 6 and the hard instance defined in (92) in Appendix B.3.

ACKNOWLEDGMENTS

We thank Adam Bouland, Tudor Giurgica-Tiron, and Aaron Sidford for helpful discussions. CZ was supported in part by the Shoucheng Zhang Graduate Fellowship. TL was supported by the National Natural Science Foundation of China (Grant Numbers 62372006 and 92365117), and a startup fund from Peking University.

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

## A    PROOF DETAILS OF OUR QUANTUM UPPER BOUND

First, we present the proof of Lemma 1. To prove this lemma, we would need a modified version of (Hamoudi, 2022, Theorem 3) given in Lemma 3. This circuit $\mathcal{C}$ uses the unitary $\mathcal{D}$ we prepare in

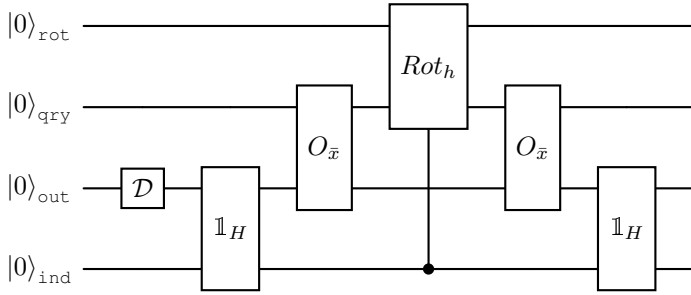

Figure 1: Circuit $\mathcal{C}$ built from $\mathcal{D}$, in which $O_{\bar{x}}$ is a quantum oracle built from $O_f$ such that $O_{\bar{x}} |i\rangle |0\rangle = |i\rangle \left| e^{f_i(\bar{x})/\epsilon'} \right\rangle$.

Algorithm 2 as a subroutine.

**Lemma 3.** *Consider two integers $1 \leq K \leq N$, a real number $\delta \in (0,1)$ and a quantum oracle $O_f$ as defined in (3). Let $\mathcal{C}$ denote a quantum circuit obtained with Algorithm 2 on input $K, \delta, O_f$ that correctly prepares the state*

$$|\phi\rangle = \sqrt{p} |\psi\rangle |0\rangle + \sqrt{1-p} |\psi_\perp\rangle |1\rangle , \tag{24}$$

*where $|\psi\rangle = \sum_i e^{f_i(\bar{x})/2\epsilon'} / \sqrt{\sum_{j \in [N]} e^{f_j(\bar{x})/\epsilon'}} |i\rangle$, $p \geq K/N$, and $|\psi_\perp\rangle$ is some unit state. The algorithm performs $O(\sqrt{KN} \log(1/\delta))$ queries to $O_f$. The circuit $\mathcal{C}$ performs two queries to $O_f$.*

*Proof.* Suppose that `QuantumMaximumFinding` returns a correct set $H$, which occurs with probability at least $1 - \delta$. The circuit $C$ as depicted in Figure 1 operates on four registers: `rot` and `ind` that contains a Boolean value, `qry` that contains a real number, and `out` that contains an integer $i \in [N]$. The indicator gate $\mathbb{1}_H$ flips `ind` when `out` contains $i \notin H$. Observe that the final state is $\mathcal{C} |0\rangle = |0\rangle_{\text{qry}} |0\rangle_{\text{ind}} |\phi\rangle_{\text{out,rot}}$ where

$$|\phi\rangle_{\text{out,rot}} = \sqrt{\mathcal{W}/\mathcal{Z}} |\psi\rangle_{\text{out}} |0\rangle_{\text{rot}} + \sqrt{1 - \mathcal{W}/\mathcal{Z}} |\psi_\perp\rangle_{\text{out}} |0\rangle_{\text{rot}} \tag{25}$$

in which $\mathcal{W} = \sum_i \exp(f_i(\bar{x})/\epsilon')$ and $|\psi_\perp\rangle$ is some state whose value is irrelevant. To bound the coefficient $p = \sqrt{\mathcal{W}/\mathcal{Z}}$, notice that by the definition of $h$, we must have $\exp(h/\epsilon') \leq \mathcal{W}/K$, or else $\sum_i \exp(f_i(\bar{x})/\epsilon') > \mathcal{W}$, which is a contradiction. Consequently, we obtain

$$p^{-1} = \frac{\mathcal{Z}}{\mathcal{W}} = \frac{(N-K) \exp(h/\epsilon') + \sum_{i \in H} \exp(f_i(\bar{x})/\epsilon')}{\sum_i \exp(f_i(\bar{x})/\epsilon')} \leq \frac{N-K}{K} + 1 = \frac{N}{K} \tag{26}$$

indicating that $p \geq \frac{K}{N}$.

Next, we bound the query complexity of the preparing of $\mathcal{C}$. Notice that for `QuantumMaximumFinding` to succeed with probability at least $1 - \delta$, it needs to make $O(\sqrt{KN} \log(1/\delta))$ queries to the oracle $O_f$ defined in (3), according to Proposition 1. Since all the remaining steps (Line 2-Line 4 in Algorithm 2) involve only classical computation, no additional quantum queries are required, the total number of quantum queries needed is $O(\sqrt{KN} \log(1/\delta))$.    □

Next, we present the proof of Lemma 1.

*Proof of Lemma 1.* The result follows directly from Lemma 3 and Proposition 3. Given that $p \geq \sqrt{K/N}$, `AmplitudeAmplification` requires only $O(\sqrt{N/K})$ calls to $\mathcal{C}$. Notice that each call to $\mathcal{C}$ only costs two quantum calls, the cost for preparing $K$ copied states is $O(\sqrt{NK})$. Combining these costs with the cost for preparing $\mathcal{C}$ gives the result of Lemma 1.    □

Note that preparing $\mathcal{D}$, we keep only the first $K$ largest elements in the distribution to make sure that the coefficient $\sqrt{p}$ before $|\psi\rangle$ can be lower bounded. This ensures that the amplitude amplification algorithm can prepare $K$ identical target states with the lowest total cost.

Next, we provide some classic results regarding $F_{\max}$ and $F_{\mathrm{smax},\epsilon}$.

**Lemma 4.** *Given $F_{\max}$ and $F_{\mathrm{smax},\epsilon}$ as defined in (12). If $x_\star$ is an approximate minimizing point of $F_{\mathrm{smax},\epsilon}$ with suboptimality $\epsilon/2$, then $x_\star$ is also an approximate minimizing point of $F_{\max}$ with suboptimality $\epsilon$.*

*Proof.* The proof is straightforward given the fact that $0 \leq F_{\mathrm{smax},\epsilon}(x) - F_{\max}(x) \leq \epsilon/2$ (see Carmon et al. 2020b, Lemma 45). Thus, we have

$$F_{\max}(x_\star) - \min_{x \in \mathbb{R}^d} F_{\max}(x_\star) = F_{\max}(x_\star) - \min_{x \in B_R(x_0)} F_{\max}(x_\star) \tag{27}$$

$$\leq F_{\mathrm{smax},\epsilon}(x_\star) - \min_{x \in B_R(x_0)} F_{\mathrm{smax},\epsilon}(x) + \frac{\epsilon}{2} \leq \epsilon. \tag{28}$$

$\square$

**Lemma 5** (Rephrased from Carmon et al. 2021, Lemma 1). *Let $f_1, \ldots, f_N$ each be $L_f$-Lipschitz and $L_g$-smooth gradients. For any $c > 0$, $r \leq c\epsilon'/L_f$, and $\lambda \leq cL_f/r$ let $C = (1 + c + c^2)e^{c+c^2/2}$. The exponentiated softmax $\Gamma_{\epsilon,\lambda}$ satisfies the following properties for any $\bar{x} \in \mathbb{R}^d$.*

- *$F_{\mathrm{smax},\epsilon}^\lambda$ and $\Gamma_{\epsilon,\lambda}$ have the same minimizer $x$ in $B_r(\bar{x})$. Moreover, for every $x \in B_r(\bar{x})$,*

$$F_{\mathrm{smax},\epsilon}^\lambda(x) - F_{\mathrm{smax},\epsilon}^\lambda(x_\star) \leq C(\Gamma_{\epsilon,\lambda}(x) - \Gamma_{\epsilon,\lambda}(x_\star)). \tag{29}$$

- *Restricted to $B_r(\bar{x})$, $\gamma_i(x) = \epsilon' e^{f_i^\lambda(x) - f_i^\lambda(\bar{x})}$ is $CL_f$-Lipshitz, $\lambda/C$ strongly convex, and $C(L_g + \lambda + L_f^2\epsilon')$-smooth.*

## B  PROOF DETAILS OF OUR QUANTUM LOWER BOUND

### B.1  WARMUP: QUANTUM LOWER BOUND FOR MULTI-ROUNDS UNSTRUCTURED SEARCH PROBLEM

First, we consider where the difficulty lies in minimizing the hard instance. Consider an algorithm that tries to achieve $\mathrm{prog}_\alpha(U^\top x) = T$ on the "shuffled" version of $f$: $\forall x, \tilde{f}(x) = \max \tilde{f}_i(U^\top x), \tilde{f}_i = f_{\Pi^{-1}(i)}$ in which $U$ is a random rotation and $\Pi$ is a random permutation.

**Definition 3.** *Based on an $\alpha$-robust $N$-element zero-chain, we define its shuffled version $\tilde{f}(x) = \max \tilde{f}_i(x)$ such that for all $x \in \mathbb{B}_R(0)$, all $y$ in a neighborhood of $x$, and all $i \in [N]$, we have*

$$\mathrm{prog}_\alpha(U^\top x) \leq p \implies \tilde{f}_i(y) = \begin{cases} f_{\Pi^{-1}(i)}(U^\top y_{[\leq p]}) & \Pi^{-1}(i) < p+1 \\ f_{\Pi^{-1}(i)}(U^\top y_{[\leq p+1]}) & \Pi^{-1}(i) = p+1 \\ f_N(U^\top y_{[\leq p]}) & \Pi^{-1}(i) > p+1, \end{cases} \tag{30}$$

*in which $U$ is an arbitrary rotation and $\Pi$ is a permutation over $[N]$.*

In order to make progress, the algorithm can either guess randomly (succeeds with minor probability) or use the oracle information. However, at first all of the gradients are $0$ unless the algorithm queries $f_{\Pi(1)}$. Then, $\Pi(2)$ needs to be discovered to uncover a new gradient direction. Thus, the algorithm needs to sequentially discover $\Pi(2), \ldots \Pi(T)$ in order to find the $x$ with progress $T$, each of which is approximately an unstructured search problem requiring a certain number of oracle queries.

From the structure of the hard instance, we see that the key of finding the minimum of an $N$-element zero chain is to find an $x$ that has a large $\mathrm{prog}_\alpha$ and an $i$ that is in the right direction (since on any other direction the oracle would only reveal information regarding the "shallower" $f_i$'s). Notice that this is essentially a multi-round version of the unstructured search problem where one need to sequentially search the $x_p, i_p$'s such that $\mathrm{prog}_\alpha(x_p) = p$ and $i_p = \Pi(p+1)$. Thus, to establish a quantum query lower bound for our problem, we begin by introducing the following variant of an unstructured search problem that has the same underlying structure as the Definition 3.

**Problem 1** (Multi-rounds unstructured search problem). *For some $N, K \geq 0$ and $d \geq 10K \ln N$, suppose there is a sequence of $K$ numbers $a_1, \ldots, a_K \in [N]$ and each number $a_i$ is associated with a different "key" $s_i$ which is a $d$-bit string. Suppose $s_1 = 0^d$. Define the search function $F_{\text{search}} \colon [N] \times \{0, 1\}^d \to \{0, 1\}^d$ to be*

$$F_{\text{search}}(a, s) = \begin{cases} s_{i+1}, & \text{if } \exists i \in [K] \text{ such that } a = a_i \text{ and } s = s_i, \\ 0^d, & \text{otherwise.} \end{cases} \tag{31}$$

*Intuitively, we can discover the $i + 1^{th}$ key if and only if we know the $i^{th}$ number $a_i$ and its corresponding key $s_i$. Suppose we have access to the following quantum oracle $O_{\text{search}}$ encoding the value of $F_{\text{search}}$:*

$$O_{\text{search}} \ket{s} \ket{a} \ket{0} \to \ket{s} \ket{a} \ket{F_{\text{search}}(a, s)}, \tag{32}$$

*the goal is to output $s_K$.*

We prove a quantum query lower bound for Problem 1 by first dividing the oracle $O_{\text{search}}$ into a series of oracles. In particular, we define a set of functions $F_{\text{search}}^{(1)}, \ldots, F_{\text{search}}^{(K)}$ where each $F_{\text{search}}^{(i)}$ is defined as

$$F_{\text{search}}^{(i)}(a, s) = \begin{cases} s_{i+1}, & \text{if } a = a_i \text{ and } s = s_i, \\ 0^d, & \text{otherwise.} \end{cases} \tag{33}$$

As we can see, each $F_{\text{search}}^{(i)}$ represents $F_{\text{search}}$ in certain input domain. Moreover, we define a series of quantum oracles $O_{\text{search}}^{(1)}, \ldots, O_{\text{search}}^{(K)}$ encoding the values of $F_{\text{search}}^{(1)}, \ldots, F_{\text{search}}^{(K)}$,

$$O_{\text{search}}^{(i)} \ket{s} \ket{a} \ket{0} \to \ket{s} \ket{a} \left| F_{\text{search}}^{(i)}(a, s) \right\rangle. \tag{34}$$

Then, one query to the oracle $O_{\text{search}}$ can be implemented using at most $K$ queries to the oracles $O_{\text{search}}^{(1)}, \ldots, O_{\text{search}}^{(K)}$. The following lemma shows that, if the $i^{th}$ key $s_i$ is not fully discovered, it would be hard to further discover the next key $s_{i+1}$ even given access to the oracle $O_{\text{search}}^{(i)}$.

**Lemma 6.** *For any quantum algorithm $\mathcal{A}$ making at most $\mathcal{T} = \sqrt{N}/6$ queries to the oracles $\{O_{\text{search}}^{(i)}\}$*

$$\mathcal{A} = O_{\text{search}}^{(i_{\mathcal{T}-1})} V_{\mathcal{T}-1} O_{\text{search}}^{(i_{\mathcal{T}-2})} \cdots O_{\text{search}}^{(i_0)} V_0, \tag{35}$$

*for any $0 \leq k < K$, if at any step $t$ of the algorithm the intermediate quantum state $\ket{\Psi_t}$ has in expectation at most $\delta$ overlap with $\ket{s_k}$, i.e., $\mathbb{E}_{s_k} |\langle s_k | \Psi_t \rangle| \leq \delta$, we have*

$$\mathbb{E}_{a_k, s_k, s_{k+1}} |\langle s_{k+1} | \Psi_t \rangle| \leq \frac{\delta}{3} + \frac{1}{2^{d/2}}. \tag{36}$$

*Proof.* For any fixed value of $a_k$ and $s_{k+1}$, we use $\ket{\Psi_t^k}$ to denote the quantum state at stage $t$. Similarly, we use $\ket{\Psi_t'}$ to denote the quantum state at stage $t$ if any query to $O_{\text{search}}^{(k)}$ is replaced by an identity operation, which can be expressed as

$$\ket{\Psi_t'} = |\langle s_k | \Psi_t' \rangle| \ket{s_k} \ket{\phi_t'} + \sqrt{1 - |\langle s_k | \Psi_t' \rangle|^2} \ket{\phi_t'^\perp}, \tag{37}$$

where $\ket{\phi_t'}$ represents the quantum state in the second and the third register that can be written as

$$\ket{\phi_t'} = \sum_{i \in [N]} \beta_{t,i} \ket{i} \ket{r_{i,t}}, \tag{38}$$

where we reorganize the state with respect to the second register, whose computational basis corresponds to all possible values of the $N$ items.

Then for any value of $a_k$, we have

$$\left\| \ket{\Psi_t^k} - \ket{\Psi_t'} \right\| \leq 2|\langle s_k | \Psi_t \rangle| \sum_{\tau=0}^{t-1} |\beta_{t, a_k}|. \tag{39}$$

Consider taking expectation over $s_k$ and $a_k$, based on our assumption that $\mathbb{E}_{s_k}|\langle s_k|\Psi_t\rangle| \leq \delta$, we have

$$\mathbb{E}_{s_k}\left\|\left|\Psi_t^k\right\rangle - \left|\Psi_t'\right\rangle\right\| \leq 2\delta \sum_{\tau=0}^{t-1}|\beta_{t,a_k}|, \tag{40}$$

and

$$\mathbb{E}_{s_k,a_k}\left\|\left|\Psi_t^k\right\rangle - \left|\Psi_t'\right\rangle\right\| \leq 2\delta \sum_{\tau=0}^{t-1}\mathbb{E}_{a_k}|\beta_{t,a_k}| \leq \frac{2\delta t}{\sqrt{N}} \leq \frac{\delta}{3}. \tag{41}$$

Since $|\Psi_t'\rangle$ is irrelavant from $s_{k+1}$, we can derive that

$$\mathbb{E}_{s_{k+1}}|\langle s_{k+1}|\Psi_t'\rangle| \leq 2^{-d/2}, \tag{42}$$

which leads to

$$\mathbb{E}_{a_k,s_k,s_{k+1}}|\langle s_{k+1}|\Psi_t\rangle| \leq \mathbb{E}_{s_k,a_k}\left\|\left|\Psi_t^k\right\rangle - \left|\Psi_t'\right\rangle\right\| + \mathbb{E}_{s_{k+1}}|\langle s_{k+1}|\Psi_t'\rangle| \leq \frac{\delta}{3} + \frac{1}{2^{d/2}}. \tag{43}$$

$\square$

**Lemma 7.** *For any $K \leq 2d$, any quantum algorithm making less than $\frac{\sqrt{N}}{6K}$ queries to the oracle $O_{\text{search}}$ defined in (32), there exists an instance of Problem 1 such that the algorithm fails to solve Problem 1 with expected probability $1 - 2^{-K}$ among all possible $s_K$, or equivalently, for any intermediate state $|\Psi\rangle$ of the algorithm, we have*

$$\mathbb{E}_{s_K}|\langle \Psi|s_K\rangle| \leq 2^{-K}. \tag{44}$$

*Proof.* We first prove a quantum query lower bound for Problem 1 with access to the oracles $O_{\text{search}}^{(1)}, \ldots, O_{\text{search}}^{(K)}$. In particular, we consider any quantum algorithm $\mathcal{A}$ for Problem 1 using $\mathcal{T} = \sqrt{N}/6$ queries to $O_{\text{search}}^{(1)}, \ldots, O_{\text{search}}^{(K)}$ for some large enough constant $C \geq 1$, it can be expressed as

$$O_{\text{search}}^{(i_{\mathcal{T}-1})}V_{\mathcal{T}-1}O_{\text{search}}^{(i_{\mathcal{T}-2})}\cdots O_{\text{search}}^{(i_0)}V_0|0\rangle. \tag{45}$$

Denote $|\psi_{\text{out}}\rangle$ to be its output quantum state. We use $\delta_i$ to denote the expected overlap with $|s_i\rangle$ among any intermediate stage of the algorithm. Then by Lemma 6, we have

$$\delta_{i+1} \leq \frac{\delta_i}{3} + \frac{1}{2^{d/2}}, \tag{46}$$

which leads to

$$\mathbb{E}_{s_K}|\langle \Psi|s_K\rangle| \leq \frac{1}{3^K} + \frac{1}{2^{d/2-1}} \leq \frac{1}{2^K}. \tag{47}$$

That is to say, any quantum algorithm making at most $\sqrt{N}/6$ queries to the oracles $O_{\text{search}}^{(1)}, \ldots, O_{\text{search}}^{(K)}$ will fail to find $s_K$ with probability at least $1 - 2^{-K}$. Since the oracle $O_{\text{search}}$ can be implemented using $K$ queries to $O_{\text{search}}^{(1)}, \ldots, O_{\text{search}}^{(K)}$, we can conclude that for any quantum algorithm making at most $\sqrt{N}/(6K)$ queries to $O_{\text{search}}$, it fails to solve Problem 1 with expected probability $1 - 2^{-K}$ among all possible $s_K$. $\square$

This result shows that quantum algorithms cannot do much better than classic algorithm when tackling a sequential structure where each step requires finding a new secret for the next direction. The only speedup in this case is the quadratic speedup brought by Grover's algorithm on the unstructured search problem. Moreover, the error of any quantum algorithm is exponential with the depth of the problem.

## B.2 Proof of Proposition 6

From an oracle $O_f$, we defined its "shuffled" version $O_{\tilde{f}}$ for $\tilde{f}$ (Definition 3) where

$$O_{\tilde{f}} |i\rangle |x\rangle = O_f \left|\Pi^{-1}(i)\right\rangle \left|U^\top x\right\rangle \tag{48}$$

in which $\Pi$ is a randomly sampled permutation over $[N]$ and $U$ is a randomly sampled rotation. According to the property of the zero chain, the subspace that we are interested in is the one in which the progress is greater than expected. Formally speaking, we define

$$P_t^\perp = \{\mathbf{x} \in B_R(0) | \exists i > t. \langle \mathbf{x}, \mathbf{u}_i \rangle > \alpha, i \in [T]\} \qquad P_t^\| = B_R(0) - P_t^\perp \tag{49}$$

To prove Proposition 6, we begin by observing that making progress on an $N$-element zero-chain is essentially solving the multi-round unstructured search problem (Problem 1). In particular, $O_{\tilde{f}}$ is the analog of $O_{\text{search}}$ defined in (32), the $p$-th function after $p_i$ is the analog of the solution of the $p$-th round of the unstructured search problem, and the informative region of this function is the analog of the "key" in this round. Moreover, we define the following "truncated" oracle $O_{\tilde{f}}^{(i)}$ of $O_{\tilde{f}}$, to be the analog of the oracle $O_{\text{search}}^{(i)}$ in Section B.1.

$$O_{\tilde{f}}^{(i)} |j\rangle |x\rangle = O_{f_i} \left|\Pi^{-1}(j)\right\rangle \left|U^\top x\right\rangle, \tag{50}$$

where $O_{f_i}$ is the evaluation oracle for the $i$-th function $f_i$. Based on this oracle, we prove the following result that is an analog of Lemma 6.

**Lemma 8.** *For any quantum algorithm $\mathcal{A}$ making at most $\mathcal{T} = \sqrt{N}/6$ queries to the oracles $\{O_{\tilde{f}}^{(i)}\}$*

$$\mathcal{A} = O_{\tilde{f}}^{(i_{\mathcal{T}-1})} V_{\mathcal{T}-1} O_{\tilde{f}}^{(i_{\mathcal{T}-2})} \cdots O_{\tilde{f}}^{(i_0)} V_0, \tag{51}$$

*for any $0 \le k < K$, if at any step $t$ of the algorithm the intermediate quantum state $|\Psi_t\rangle$ satisfies*

$$\mathop{\mathbb{E}}_{\Pi, U} \|\mathcal{P}(\Pi^{-1}(k)) |\Psi_t\rangle\| \le \delta, \tag{52}$$

*where $\mathcal{P}(j)$ denotes the projector onto the informative region of the $j$-th function $f_j$, we have*

$$\mathop{\mathbb{E}}_{\Pi, U} \|\mathcal{P}(\Pi^{-1}(k+1)) |\Psi_t\rangle\| \le \frac{\delta}{3} + 2e^{-d\alpha^2/2}. \tag{53}$$

*Proof.* We use $\left|\Psi_t^k\right\rangle$ to denote the quantum state at stage $t$. Similarly, we use $|\Psi_t'\rangle$ to denote the quantum state at stage $t$ if any query to $O_{\tilde{f}}^{(k)}$ is replaced by an identity operation, which can be expressed as

$$|\Psi_t'\rangle = \|\mathcal{P}(\Pi^{-1}(k)) |\Psi_t'\rangle \| \mathcal{P}(\Pi^{-1}(k)) |\Psi_t'\rangle + \sqrt{1 - \|\mathcal{P}(\Pi^{-1}(k)) |\Psi_t'\rangle\|^2} |\perp\rangle, \tag{54}$$

where $\mathcal{P}(\Pi^{-1}(k)) |\Psi_t'\rangle$ represents the quantum state in the second and the third register that can be written as

$$\mathcal{P}(\Pi^{-1}(k)) |\Psi_t'\rangle = \sum_{i \in [N]} \beta_{t,i} |i\rangle |x_{i,t}\rangle, \tag{55}$$

where we reorganize the state with respect to the second register, whose computational basis corresponds to all possible values in $[N]$. Then we can derive that

$$\left\| \left|\Psi_t^k\right\rangle - |\Psi_t'\rangle \right\| \le 2\|\mathcal{P}(\Pi^{-1}(k)) |\Psi_t'\rangle\| \sum_{\tau=0}^{t-1} |\beta_{\tau,x}|. \tag{56}$$

Consider taking expectation over $\Pi$ and $U$, based on our assumption that $\mathop{\mathbb{E}}_{\Pi, U} \|\mathcal{P}(\Pi^{-1}(k)) |\Psi_t\rangle\| \le \delta$, we have

$$\mathop{\mathbb{E}}_{s_k} \left\| \left|\Psi_t^k\right\rangle - |\Psi_t'\rangle \right\| \le 2\delta \sum_{\tau=0}^{t-1} |\beta_{\tau,\Pi^{-1}(k)}|, \tag{57}$$

and

$$\mathop{\mathbb{E}}_{\Pi,U} \left\| |\Psi_t^k\rangle - |\Psi_t'\rangle \right\| \le 2\delta \sum_{\tau=0}^{t-1} \mathbb{E}|\beta_{\tau,\Pi^{-1}(k)}| \le \frac{2\delta t}{\sqrt{N}} \le \frac{\delta}{3}. \tag{58}$$

Since $|\Psi_t'\rangle$ is irrelavant from $U$ and $\Pi$, using a known fact in the probability theory (see e.g., Zhang & Li 2023, Lemma 17)

$$\Pr_{\mathbf{u}}(\langle x, \mathbf{u}\rangle \ge \alpha) \le 2e^{-d\alpha^2/2}, \tag{59}$$

we have

$$\mathop{\mathbb{E}}_{\Pi,U} \left\| \left\| \mathcal{P}(\Pi^{-1}(k)) |\Psi_t'\rangle \right\| \right\| \le 2e^{-d\alpha^2/2}, \tag{60}$$

by which we can conclude that

$$\mathbb{E}\|\mathcal{P}(\Pi^{-1}(k+1)) |\Psi_t\rangle\| \le \mathop{\mathbb{E}}_{\Pi,U} \left\| |\Psi_t^k\rangle - |\Psi_t'\rangle \right\| + \mathop{\mathbb{E}}_{\Pi,U} \|\|\mathcal{P}(\Pi^{-1}(k)) |\Psi_t'\rangle\| \tag{61}$$

$$\le \frac{\delta}{3} + 2e^{-d\alpha^2/2}. \tag{62}$$

$\square$

The next lemma is an analog of Lemma 7.

**Lemma 9.** *For any $K \le d\alpha^2/4$, any quantum algorithm making less than $\frac{\sqrt{N}}{6K}$ queries to the oracle $O_{\tilde{f}}$ defined in (32), for any intermediate state $|\Psi\rangle$ of the algorithm, we have*

$$\mathop{\mathbb{E}}_{\Pi,U} \|\mathcal{P}(\Pi^{-1}(K)) |\Psi\rangle\| \le 2^{-K}. \tag{63}$$

*Proof.* We first consider the case with access to the oracles $O_{\tilde{f}}^{(1)}, \ldots, O_{\tilde{f}}^{(K)}$. In particular, we consider any quantum algorithm $\mathcal{A}$ for Problem 1 using $\mathcal{T} = \sqrt{N}/6$ queries to $O_{\tilde{f}}^{(1)}, \ldots, O_{\tilde{f}}^{(K)}$ for some large enough constant $C \ge 1$, it can be expressed as

$$O_{\tilde{f}}^{(i_{\mathcal{T}-1})} V_{\mathcal{T}-1} O_{\tilde{f}}^{(i_{\mathcal{T}-2})} \cdots O_{\tilde{f}}^{(i_0)} V_0 |0\rangle. \tag{64}$$

Denote $|\psi_{\text{out}}\rangle$ to be its output quantum state. We use $\delta_i$ to denote the expected value of $\|\mathcal{P}(\Pi^{-1}(i)) |\Psi\rangle\|$ among any intermediate stage of the algorithm. Then by Lemma 6, we have

$$\delta_{i+1} \le \frac{\delta_i}{3} + 2e^{-d\alpha^2/2}, \tag{65}$$

which leads to

$$\delta_K \le \frac{1}{3^K} + 4e^{-d\alpha^2/2} \le \frac{1}{2^K}. \tag{66}$$

That is to say, any quantum algorithm making at most $\sqrt{N}/6$ queries to the oracles $O_{\tilde{f}}^{(1)}, \ldots, O_{\tilde{f}}^{(K)}$ will fail to find $s_K$ with probability at least $1 - 2^{-K}$. Since the oracle $O_{\tilde{f}}$ can be implemented using $K$ queries to $O_{\tilde{f}}^{(1)}, \ldots, O_{\tilde{f}}^{(K)}$, we can conclude that for any quantum algorithm making at most $\sqrt{N}/(6K)$ queries to $O_{\tilde{f}}$, any intermediate state $|\Psi\rangle$ of the algorithm satisfies (63). $\square$

**Lemma 10.** *Consider the unitary $A_n$ with $n$ queries to $O_{\tilde{f}}$ of the form*

$$A_n = O_{\tilde{f}} V_{n-1} O_{\tilde{f}} \cdots O_{\tilde{f}} V_0 \tag{67}$$

*in which $n = \sqrt{N}/(6K)$ and $K = 4\log T + \frac{1}{2}\log N + 5$. Then for any input state $|\psi\rangle$, we have*

$$\gamma(n) = \mathbb{E}\left[ \left\| P_K^\perp A_n |\psi\rangle \right\|^2 \right] \le \frac{1}{16\sqrt{N}T^4}, \tag{68}$$

*as long as the dimension $d \ge 4K/\alpha^2$.*

*Proof.* Observe that we can achieve $\gamma(n)$ by the summation of overlap given in Lemma 9 and an upper bound of the success probability of random guesses. The first term is at most

$$2^{-K} = \frac{1}{32\sqrt{N}T^4}, \tag{69}$$

by Lemma 9. Meanwhile, the second term is at most

$$T \cdot 2e^{-d\alpha^2/2} \leq \frac{1}{32\sqrt{N}T^4}, \tag{70}$$

by which we can conclude that

$$\gamma(n) \leq \frac{1}{16\sqrt{N}T^4}. \tag{71}$$

$\square$

Next, we state the fact that it is fundamentally hard to make progress by random guessing.

**Lemma 11** (Cannot make progress through guessing). *Let $\{\mathbf{u}_i\}_{i=1}^t$ be a set of fixed orthonormal vectors. Choose $\{\mathbf{u}_i\}_{i=t+1}^T$ uniformly randomly such that $\{\mathbf{u}_i\}_{i=1}^T$ is orthonormal. Then*

$$\forall \mathbf{x} \in B_R(0), \Pr_{\mathbf{u}_{t+1},\ldots\mathbf{u}_T}(\text{prog}_{\alpha_T} \mathbf{x} \geq t) \leq 2Te^{-(d-T)/32T^3} \tag{72}$$

*in which $\alpha_T = 1/(4T^{3/2})$.*

*Proof.* We use a standard union bound with some common facts to prove this lemma. First, using a union bound we have

$$\Pr_{\mathbf{u}_{t+1},\ldots\mathbf{u}_T}(\text{prog}_\alpha(x) \geq t) \leq \sum_{i=t+1}^T \Pr(\langle x, \mathbf{u}_i \rangle \geq \alpha) \tag{73}$$

$$= (T-t)\Pr(\langle x, \mathbf{u}_i \rangle \geq \alpha) \tag{74}$$

where the final equality is due to the symmetry among $\mathbf{u}_i$'s. Next, using a known fact in the probability theory (see e.g., Zhang & Li 2023, Lemma 17) we have

$$\Pr_{\mathbf{u}}(\langle x, \mathbf{u} \rangle \geq c) \leq 2e^{-dc^2/2}. \tag{75}$$

Considering that $\mathbf{u}_{t+1},\ldots\mathbf{u}_T$ are sampled from a $d-T$ dimention space orthogonal to the fixed vectors, we have

$$\Pr_{\mathbf{u}_{t+1},\ldots\mathbf{u}_T}(\text{prog}_\alpha(x) \geq t) \leq 2Te^{-(d-T)/32T^3}. \tag{76}$$

$\square$

With these results at hand, we now articulate the proof technique for Proposition 6. The proof technique involves using a quantum hybrid such that it is almost impossible for the last hybrid to find the minimizing point. First, we introduce a truncated oracle

$$\tilde{O}_t |i\rangle |\mathbf{x}\rangle |y\rangle \rightarrow |i\rangle |\mathbf{x}\rangle \left|y \oplus f_{\Pi^{-1}(i)}\left(u_1^\top \mathbf{x}, u_2^\top \mathbf{x} \ldots u_t^\top \mathbf{x}, 0, \ldots, 0\right)\right\rangle. \tag{77}$$

Note that $\tilde{O}_t$ does not reveal any information about $U_{>t}$. Given a quantum algorithm $\mathcal{A}$ making $n \leq T\sqrt{N}/CK$ queries, the algorithm can be represented in the form

$$\mathcal{A} = V_n O_{\tilde{f}} V_{n-1} \cdots O_{\tilde{f}} V_0. \tag{78}$$

From this algorithm we could build $k$ hybrids:

$$
\begin{aligned}
A_0 &= \mathcal{A} = V_{k\sqrt{N}/CK} O_{\tilde{f}} \cdots \tilde{O}_{10\log T} V_{\sqrt{N}/CK-1} \cdots O_{\tilde{f}} V_1 O_{\tilde{f}} V_0, \\
A_1 &= V_{k\sqrt{N}/CK} O_{\tilde{f}} \cdots \tilde{O}_{10\log T} V_{\sqrt{N}/CK-1} \cdots \tilde{O}_{10\log T} V_1 \tilde{O}_{10\log T} V_0, \\
&\cdots \\
A_k &= V_{k\sqrt{N}/CK} \tilde{O}_{10k\log T} \cdots \tilde{O}_{10\log T} V_{\sqrt{N}/CK-1} \cdots \tilde{O}_{10\log T} V_1 \tilde{O}_{10\log T} V_0.
\end{aligned}
\tag{79}
$$

Note that the difference between two consecutive hybrids $A_{i-1}$ and $A_i$ is that $\sqrt{N}/CK$ of the oracles $O_{\tilde{f}}$ are changed to $\tilde{O}_{t \log T}$. Following the standard hybrid argument, in the subsequent lemmas we demonstrate that the output of the sequence of unitaries (the difference between $A_t$ and $A_{t-1}$) on random input is similar.

**Lemma 12.** *For any $t \in [T-1]$ and any $n \leq \sqrt{N}/CK$ for $K = 4 \log T + \frac{1}{2} \log N + 4$, consider the following two sequences of unitaries,*

$$\mathcal{A}(n) = O_{\tilde{f}} V_n O_{\tilde{f}} V_{n-1} \cdots O_{\tilde{f}} V_0 \tag{80}$$

*and*

$$\hat{\mathcal{A}}_t(n) = \tilde{O}_t V_n \tilde{O}_t V_{n-1} \cdots \tilde{O}_t V_0. \tag{81}$$

*Given that $d \geq T + \max(32T^3 \log(32\sqrt{N}T^5), 32T^3 \log(12T))$, we have*

$$\delta(n) := \mathbb{E}\left[\left\|(\hat{\mathcal{A}}_t - \mathcal{A})\,|\psi\rangle\right\|^2\right] \leq \frac{n}{16\sqrt{N}T^4} \tag{82}$$

*Proof.* We use induction to prove this lemma. For $n = 1$, we have

$$\delta(1) = \mathbb{E}\left[\left\|(O_{\tilde{f}} - \tilde{O}_t)\,|\psi\rangle\right\|^2\right] \leq \frac{1}{16\sqrt{N}T^4} \tag{83}$$

where the inequality follows from Lemma 11. Suppose the inequality (82) holds for $\forall n \leq \tilde{n}$ for some $\tilde{n} < \sqrt{n}$. For $n = \tilde{n} + 1$, by Lemma 10, we first have that

$$\mathbb{E}\left[\left\|P_t^{\perp}\,|\psi_{\tilde{n}}\rangle\right\|^2\right] \leq \frac{1}{16\sqrt{N}T^4} \tag{84}$$

Next, we have

$$\delta(n) = \delta(1) = \mathbb{E}\left[\left\|(O_{\tilde{f}} - \tilde{O}_t)\,|\psi_{\tilde{n}}\rangle\right\|^2\right] + \delta(\tilde{n}) \tag{85}$$

$$\leq \mathbb{E}\left[\left\|P_t^{\perp}\,|\psi_{\tilde{n}}\rangle\right\|^2\right] + \delta(\tilde{n}) \leq \frac{n}{16\sqrt{N}T^4}, \tag{86}$$

in which inequality (86) is due to the fact that only $\mathbf{u}_i$ with $i > t$ can contribute to the expectation. $\square$

**Lemma 13.** *If $d \geq T + 32T^3 \log\left(32\sqrt{N}T^5\right)$, for an $N$-element zero chain $f$ we have*

$$\mathbb{E}[\|(A_t - A_{t-1})\,|0\rangle\|^2] \leq \frac{1}{16T^4}. \tag{87}$$

*Proof.* From the definition of the hybrid unitaries $A_i$ in Eq. (79), we have

$$\mathbb{E}[\|(A_i - A_{i-1})\,|0\rangle\|^2] = \mathbb{E}\left[\left\|(\mathcal{A}(\sqrt{N}/CK) - \hat{\mathcal{A}}(\sqrt{N}/CK))\,|\psi\rangle\right\|^2\right]. \tag{88}$$

Thus, according to Lemma 12 the expectation is bounded by

$$\mathbb{E}[\|(A_i - A_{i-1})\,|0\rangle\|^2] \leq \frac{1}{16T^4}. \tag{89}$$

$\square$

Equipped with Lemma 13, we are now ready to prove Proposition 6.

*Proof of Proposition 6.* First, suppose that the quantum algorithm $\mathcal{A}$ only makes at most $T\sqrt{N}/CK^2$ queries, in which $K = 4 \log T + \frac{1}{2} \log N + 4$. According to the construction of the hybrid (79), the last vector $\mathbf{u}_T$ is never revealed. Thus, by Lemma 11, letting $d \geq T + \max(32T^3 \log(32\sqrt{N}T^5), 32T^3 \log(4T/\delta))$, the last unitary in the hybrid can find an $\mathbf{x}$ with sufficient progress with probability at most $\delta/2$.

Next, we can bound the total variance distance between the distributions obtained by sampling $A_0 \left| 0 \right\rangle$ and $A_K \left| 0 \right\rangle$. By Lemma 13 and the Cauchy-Schwarz inequality, we have

$$\mathbb{E}[\|(A_K - A_0)\left| 0 \right\rangle\|^2] \leq K \sum_{i=0}^{K-1} \mathbb{E}[\|(A_i - A_{i-1})\left| 0 \right\rangle\|^2] \leq \frac{1}{16T^2}. \tag{90}$$

Now, applying Markov's inequality, we can establish that

$$\Pr\left[\|(A_K - A_0)\left| 0 \right\rangle\|^2 > \frac{1}{4T}\right] \leq \frac{1}{4T}, \tag{91}$$

which indicates that the total variance distance can be bounded by summing up $1/(4T) + 1/(4T) = 1/(2T) \leq \delta/2$, provided that $T \geq \delta^{-1}$. Combining these two parts allows us to upper bound the success probability by $\delta$. $\square$

## B.3 PROOF OF THEOREM 4

First of all, we provide the hard instance we use in the proof of the main theorem:

$$\hat{f}_i^{\{T,N,\ell\}}(x) := \begin{cases} \psi_{\alpha,\ell}\left(\frac{x_{[i]} - x_{[i-1]}}{2}\right) & i \leq T, \\ 0 & \text{otherwise}, \end{cases} \tag{92}$$

where we denote $\alpha_T = \frac{1}{4T^{3/2}}$, $x_{[0]} := \frac{1}{\sqrt{T}}$, and $\psi_{\alpha,\ell} : \mathbb{R} \to \mathbb{R}$ is defined as

$$\psi_{\alpha,\ell}(t) := \begin{cases} 0 & |t| \leq \alpha \\ \frac{\ell}{2}(t-\alpha)^2 & \alpha \leq |t| \leq \ell^{-1} + \alpha \\ |t| - \alpha - \frac{1}{2\ell} & \text{otherwise}. \end{cases}$$

The properties of this hard instance $\hat{f}^{\{T,N,\ell\}}$ is given in the following lemma.

**Lemma 14** (Carmon et al. 2021, Lemma 2)**. *For every $T, N \in \mathbb{N}$ and $\ell \geq 0$, such that $T \leq N$, we have that*

1. *The hard instance $(\hat{f}_i^{\{T,N,\ell\}})_{i \in N}$ is an $\alpha_T$-robust $N$-element zero-chain.*

2. *The function $\hat{f}_i^{\{T,N,\ell\}}$ is 1-Lipschitz and $\ell$-smooth for every $i \in [N]$.*

3. *For any $x \in \mathbb{R}^d$ with $\text{prog}_{[\alpha_T]}(x) < T$, the objective $\hat{F}^{\{T,N,\ell\}}(x) = \max_{i \in [N]} \hat{f}_i^{\{T,N,\ell\}}(x)$ satisfies*

$$\hat{F}^{\{T,N,\ell\}}(x) - \min_{x_\star \in \mathbb{B}_1(0)} \hat{F}^{\{T,N,\ell\}}(x_\star) \geq \psi_{\alpha_T,\ell}\left(\frac{3}{8T^{3/2}}\right) \geq \min\left(\frac{1}{8T^{3/2}}, \frac{\ell}{32T^3}\right).$$

Next, we present the detailed proof of Theorem 4 using Proposition 6 and the hard instance defined in (92).

*Proof of Theorem 4.* Theorem 4 is a direct result of Proposition 6 and the property of (92). Notice that from Proposition 6, when $d \geq T + \max(32T^3 \log(32\sqrt{N}T^5), 32T^3 \log(4T/\delta))$ and $T \geq 3$, the success probability of the quantum algorithm $\mathcal{A}$ making at most $T\sqrt{N}/C \cdot (4 \log T + \frac{1}{2} \log N + 4)^2$ queries discovering $\mathbf{u}_T$ is at most $1/3$. Set

$$T = \frac{1}{5} \max\left(\left(\frac{L_f R}{\epsilon}\right)^{2/3}, \left(\frac{L_g R^2}{\epsilon}\right)^{1/3}\right) \quad \text{and} \quad \mathcal{L} = L_g R/L_f. \tag{93}$$

Note that under this setting we have that the lower bound number of queries is $T\sqrt{N}/C(4 \log T + \frac{1}{2} \log N + 4)^2 = \tilde{\Omega}(\sqrt{N}\epsilon^{-2/3})$. According to the part 3 of Lemma 14, if the algorithm failed in discovering the $T^{\text{th}}$ direction $\mathbf{u}_T$ (i.e., $\text{prog}_{\alpha_T}(x) < T$), the suboptimality of the output minimizing point is at least $\min\{\frac{1}{8T^{3/2}}, \frac{\mathcal{L}}{32T^3}\} \leq \epsilon$. Thus, we have

$$\Pr\left[F_{\max}(x_{\text{out}}) - \min_{x \in B_R^d(0)} F_{\max}(x) \geq \epsilon\right] \leq \frac{1}{3}. \tag{94}$$

$\square$

