# OpenReview forum: "Near-Optimal Quantum Algorithm for Minimizing the Maximal Loss"
_ICLR.cc/2024/Conference — ICLR 2024 poster_

### Official Review · Reviewer_gKCY · 2023-11-01

**Soundness:** 3 good
**Presentation:** 3 good
**Contribution:** 3 good
**Rating:** 6
**Confidence:** 2

**Summary:**

The paper considers a convex optimization problem where the function to be minimized is the maximum of $N$ convex Lipschitz functions. The paper presents a quantum algorithm which finds an $\epsilon$-good solution using $\tilde{O}(\sqrt{N}\epsilon^{-5/3} + \epsilon^{-8/3})$ queries. The paper complements the algorithmic result with a lower bound showing that the dependence on $N$ is optimal up to logarithmic factors.

The paper builds on the classical algorithm of Carmon et. al. which is based on the Ball optimization acceleration technique, and then identifies parts where a quadratic quantum speedup can be applied.

**Strengths:**

The results of the paper are novel and interesting. Furthermore, the dependence of the number of queries of the algorithm that is presented in the paper on the number of functions $N$ is shown to be optimal up to logarithmic factors.

I also find the paper to be generally well written.

**Weaknesses:**

There is still a gap between the query-complexity of the algorithm and the shown lower bound in terms of the dependence on $\epsilon$, but this is not a significant weakness of the paper.

**Questions:**

The quadratic speedup is a recurring feature of quantum algorithms and in many cases the phenomenon at the core of the speedup is the same. The techniques described in this paper can perhaps be applied to other computational/optimization problems. I wonder if one can state a general meta-theorem of the form: "For a wide class of computational/optimization problems, if we have a classical algorithm that solves it and which satisfy some properties, then we can find a corresponding quantum algorithm with a quadratic speedup".

---

> ### Author Response · Authors · 2023-11-18
>
> We appreciate your positive feedback and detailed suggestions!
>
> Please refer to our general response for your comment on the gap between our upper bound and lower bound.
>
> Regarding your question on a potential meta-theorem, we agree that it is indeed a good research question to ask.
> There have been many quantum algorithms on important subroutines adopted in many classical algorithms.
> For example, our paper exploits a quantum speedup for multiple sampling from the same distribution found by Hamoudi [1].
> However, the possibility of deriving a meta-theorem appears uncertain to us. Distinct quantum approaches and techniques, albeit that the source of the speedup might be the same, are needed for different scenarios with different settings.
> In particular, conceptually Gao et al. [2] and Bouland et al. [3] both achieve speedup on sampling, but the sampling tasks are slightly different, leading to entirely different algorithms compared to the one used in our algorithm.
> Generally, we think it is nontrivial to summarize types of sampling tasks with similar quantum speedup due to the diversity and specificity of scenarios.
>
> **References**
>
> [1] Yassine Hamoudi. "Preparing Many Copies of a Quantum State in the Black-Box Model." In Phys. Rev. A, vol. 105, no. 6, p. 062440, Jun. 2022, doi: 10.1103/PhysRevA.105.062440. arXiv:2207.11014
>
> [2] Minbo Gao, Zhengfeng Ji, Tongyang Li, and Qisheng Wang. "Logarithmic-Regret Quantum Learning Algorithms for Zero-Sum Games." To appear in the Conference on Neural Information Processing Systems, 2023. arXiv:2304.14197
>
> [3] Adam Bouland, Yosheb M. Getachew, Yujia Jin, Aaron Sidford, and Kevin Tian. "Quantum speedups for zero-sum games via improved dynamic Gibbs sampling." In International Conference on Machine Learning, pp. 2932-2952. PMLR, 2023. arXiv: 2301.03763

---

> > ### Comment · Reviewer_gKCY · 2023-12-02
> > **Reply to authors**
> >
> > I would like to thank the authors for their reply,
> >
> > I'm still positive regarding the submitted work and I retain my review.

---

### Official Review · Reviewer_ixEA · 2023-11-04

**Soundness:** 3 good
**Presentation:** 3 good
**Contribution:** 3 good
**Rating:** 8
**Confidence:** 3

**Summary:**

The paper studies the question of minimizing the maximum of N convex Lipschitz loss functions f1, f2,...,fN. It is known that you can do so with O(N eps^{-2/3} + eps^{-8/3}) queries. The paper here studies a quantum version of the problem and achieve a complexity that is sqrt{N} eps^{-5/3} + eps^{-8/3}. While the eps-dependence is slightly worse, the dependence on N has improved by a quadratic factor.

The main idea is to take a classical algorithm for the problem and modify one of the key-steps to exploit quantum advantage. Specifically, the authors exploit the fact that the classical algorithm uses a regularized ball optimization oracle (BROO) which is similar to a Gibbs sampling problem. The latter is known to have a quantum advantage and exploiting the better algorithms for that gives the final result.

**Strengths:**

The paper studies a natural problem in optimization and shows that quantum algorithms could do better.

**Weaknesses:**

A disadvantage is that the algorithm in a sense takes an existing classical algorithm and replaces a step in it with a suitable, known, quantum algorithm. So novelty is a bit low.

**Questions:**

None.

---

> ### Author Response · Authors · 2023-11-18
>
> We appreciate your positive feedback and detailed comments!
>
> Please refer to our general response for your comment on the novelty of our algorithm.

---

### Official Review · Reviewer_J3o2 · 2023-11-06

**Soundness:** 3 good
**Presentation:** 3 good
**Contribution:** 2 fair
**Rating:** 5
**Confidence:** 3

**Summary:**

This paper designs new quantum algorithm for the problem of minimizing the maximum of $N$ convex functions with near optimal quantum query complexity.

Given $f_1,\ldots, f_N$ convex and $L$-Lipschitz functions from $\mathbb{R}^d$ to $\mathbb{R}$. The function $F_{max}(x)$ is defined as:  $F_{max}(x)$  = ${max}_{i} \\{f_i(x)\\}$.

The goal is to find an $x*$ so that $F_{max}(x*) - inf_{x} F_{max}(x) \leq \epsilon$.

This is a fundamental optimization task and substantial research has gone in identifying the number of oracle queries required (say to $f_i$s and its gradients - called the first order oracle) as a function of both $N$ -- the number of functions, and $\epsilon$ -- the error parameter. Tight results have been established (matching upper and lower bounds in the number of queries) in the literature for some regimes of parameters (especially for $\epsilon \geq 1/\sqrt{N}$).  However, these prior research only focused on 'classical queries.' In this paper the authors consider 'quantum query complexity' of the problem.  It is well established that in general quantum query algorithms can get a quadratic speed up over classical query algorithm. The present paper extends that to the above described minimizing-the-maximum-loss problem.

The paper establishes an upper bound of $\tilde{O}(\sqrt{N} \epsilon^{-5/3} + \epsilon^{-8/3})$ and a lower bound of $\tilde{\Omega}(\sqrt{N}\epsilon^{-2/3})$ on the number of quantum queries required for solving the problem.

Thus with respect to $N$, the established results are optimal. But there is a gap in upper and lower bounds in terms of $\epsilon$. Closing this gap, the authors pose as a natural open question. The quantum oracle model they use is what is typically seen in the literature and is called the zeroth-order oracle  in the optimization terminology.

**Strengths:**

The strength is that the paper  reports progress on a fundamental and well-studied optimization task. It is natural to consider the question whether the speed up you get in other quantum search algorithms (such as in Grover's algorithm) can be lifted to the optimization literature also.  So in that sense, the results established are clean and informative.

**Weaknesses:**

A weakness is the originality. It appears that nothing very new is going on here and the results (at least with respect to $N$) are expected. Also, it is not optimal with respect to $\epsilon$. In fact, known classical algorithms do better with respect to $\epsilon$. So in that sense the picture is not complete, which the authors do not seem to address.

**Questions:**

It will be nice to further explain  the reason for the gap in upper and lower bounds, in particular in comparison with classical algorithms. Is the reason classical algorithm does better in $\epsilon$ is because of more powerful (first order as opposed to zeroth order) oracle? Also, I am not clear what you mean by the statement "However, as far as we know, ..." the last sentence before Contributions. Probably it is good idea to make sure it is indeed open.  It is probably fine for smaller results in the paper, but for the main question you are considering, it is better to make sure that the status of the problem.

**Details Of Ethics Concerns:**

No Concern.

---

> ### Author Response · Authors · 2023-11-18
>
> We appreciate your detailed suggestions!
>
> Please refer to our general response for your comment and question on the novelty and unmatched bounds.
>
> Regarding your comment on the uncertainty in the wording of the contribution part of our paper, we have made sure that the problem is still open and changed the statement accordingly.

---

> > ### Comment · Reviewer_J3o2 · 2023-12-04
> > **Reply**
> >
> > I thank the authors for the response. As I mentioned in the original review, I like the direction of the work. However, I still feel that with the gap between the upper and lower bound in the critical parameter $\epsilon$, the paper is not above acceptance threshold. So I will retain my score.

---

### Official Review · Reviewer_aSjj · 2023-11-06

**Soundness:** 3 good
**Presentation:** 2 fair
**Contribution:** 2 fair
**Rating:** 5
**Confidence:** 3

**Summary:**

The paper presents an improved quantum algorithm for minimizing the maximal loss of $N$ convex functions, the key idea is quantum Gibbs sampling subroutine that improves the classical subroutine.

The paper studies the problem of minimizing the loss of $N$ convex functions, i.e., $\min_{x} \max_{i \in [N]}f_{i}(x)$ for convex functions $f_{1}, \ldots, f_{N}$ given gradient oracle. The best classical algorithm for finding $\epsilon$-approximate solution is given by [Carmon et al. 2021] with $O(N\epsilon^{-2/3} +\epsilon^{-8/3})$ queries, which is known to be optimal in the low accuracy regime $\epsilon \geq 1/\sqrt{N}$. The idea is to approximate the maximal loss with softmax function and use the improved ball optimization oracle [Carmon et al. 2020].

The paper studies quantum algorithms and the algorithm is equipped with zero-th order quantum oracle (it can be used to obtain gradient oracle) and improve the bound to $O(\sqrt{N}\epsilon^{-5/3} +\epsilon^{-8/3})$. It also gives a lower bound saying $\Omega(\sqrt{N}\epsilon^{-2/3})$ queries are necessary for finding $\epsilon$-approximate solution.

The main idea of the algorithm is an improved estimation of the gradient, which in turns boils down to improved quantum Gibbs sampling procedure (note the gradient draws from a distribution, which requires $N$ queries, but quantum algorithm could take advantage and only needs $\sqrt{N}$ queries)

**Strengths:**

The paper gives improved quantum algorithm for a basic problem in convex optimization. The idea of using quantum Gibbs sampling for improve is interesting.

**Weaknesses:**

The major unsatisfactory point is the unmatched upper and lower bound.

The presentation is fine in general, but it could be improved (e.g. I found the English a bit awkward some time). Some detailed comments:

(1) Page 1.  "Nesterov (2018) showed ..."  I believe it is an old result, not shown recently. Probably change it to something like "it is known that .... e.g., see Nesterov (2018) ".

(2) Page 4. "named ket" I don't think you need to name it. For quantum people, they know what's the notation means; for non-quantum people, it is not informative and it does not appear twice in the paper.

(3) Line 5 in Algorithm 1. The subscription is confusing.

(4) Page 8 " given in Ref.  Carmon et al. (2021)". Remove Ref.

**Questions:**

I tried to understand Algorithm 2 (quantum sampling algorithm), but due to some time constraints, I did not fully understand it and I believe some better explanation should make the paper better.

I understand that you first find the $K$ largest elements of the distribution, and then you set the probability of rest element equals the probability of the $K$-th element -- this changes the distribution. Do you need some rejection sampling procedure afterwards?
(Is the answer is in figure 1? Then it would be better to move figure to the main paper).

---

> ### Author Response · Authors · 2023-11-18
>
> We appreciate your detailed comments!
>
> Please refer to our general response for your comment on the unmatched upper and lower bound.
>
> Regarding your question about Algorithm 2,
> we did not use any rejection sampling for the preparation of the states.
> The unitary $\mathcal{D}$ that is built in the step 3 of our Algorithm 2 is used only as a subroutine in the circuit that is shown in Figure 1.
> With several additional gates, in Figure 1 we build a circuit that prepares a state that contains the desired and an ancillary garbage state.
> We then apply the amplitude amplification algorithm to prepare the desired state, as shown in step 5 of Algorithm 2.
> The reason we first only consider the first $K$ largest element is to guarantee that the amplitude amplification can extract the state we want efficiently: in Eq. (26) we lower bound the coefficient before the target state, guaranteeing a total cost of $\tilde{O}(\sqrt{NK})$ according to Proposition 3.
> We have made updates correspondingly in the paper to improve the presentation of Algorithm 2.
>
> We also thank you for detailed suggestions regarding presentation - they have been fixed in the revised version.

---

### Author Response · Authors · 2023-11-18

We thank all reviewers for their thoughtful consideration of our paper. We appreciate your generally positive recognition of our contributions and helpful suggestions for further improvements. Here we address some common questions that arose in multiple reviews.

Regarding the gap between the upper bound and lower bound in our paper, we would like to first note that our algorithm is easy to understand and implement.
Additionally, the algorithm is near optimal in the sense that the dependency with respect to $N$ is indeed tight.
From our understanding, the gap is not due to a weaker quantum zeroth-order oracle comparing to the classical first-order oracle since using Jordan's algorithm [1] we can still compute gradients from a quantum zeroth-order oracle. Instead, the gap is mainly due to the technique we use for the quantum speedup on sampling multiple times from the same distribution.
Conceptually, the technique is making a trade-off between $N$ and $\epsilon$.
Comparing to the classical algorithm where the sampling cost is $O(N)$, our sampling subroutine takes $\tilde{O}(\sqrt{NT})$ to prepare the samplings according to Lemma 3 in Appendix A, in which $T$ is the number of samples needed and is relevant to $\epsilon$.
This is because our quantum algorithm needs to prepare $T$ independent quantum states using amplitude amplification in Line 5 of Algorithm 2, while classical algorithm directly computes the distribution and the number of samples is irrelevant to the cost.
Although each sample only takes $\sqrt{N/T}$ queries according to Proposition 3 and Lemma 3, the repetitive preparation would cost $\tilde{O}(\sqrt{NT})$ in total.
Notably, previous theory results on quantum machine learning focused more on speedups of dimension factors, even though the dependency on $\epsilon$ is potentially of higher order. For example:

Ref. [2] in ICML 2019 addressed linear classification and kernel-based classification problem for $n$ $d$-dimensional data points with quantum gate complexity $\tilde{O}(\sqrt{n}/\epsilon^{4}+\sqrt{d}/\epsilon^{8})$, whereas classically the bound is $\tilde{O}(n/\epsilon^{2}+d/\epsilon^{2})$.

Ref. [3] in ICML 2020 improved AdaBoost for learning a concept class $\mathcal{C}$ with VC-dimension $VC(\mathcal{C})$ from $\gamma$-weak learner to complexity $\tilde{O}(\sqrt{VC(\mathcal{C})}/\gamma^{11})$ in the quantum setting, whereas classically the bound is $\tilde{O}(VC(\mathcal{C})/\gamma^{4})$.

Ref. [4] in AAAI 2021 studied general $n\times d$ matrix games with quantum gate complexity $\tilde{O}(\sqrt{n}/\epsilon^{4}+\sqrt{d}/\epsilon^{7})$, whereas classically the bound is $\tilde{O}(n/\epsilon^{2}+d/\epsilon^{2})$.

Ref. [5, 6] in ICML 2023 and NeurIPS 2023 respectively gave quantum algorithms for $m\times n$ matrix zero-sum games with complexity $\tilde{O}(\sqrt{m+n}/\epsilon^{2.5})$, whereas classically the bound is $\tilde{O}((m+n)/\epsilon^{2})$.

We agree that reducing this gap stands as an interesting open question that is worth further exploring.

**Reference**

[1] Stephen P. Jordan. "Fast quantum algorithm for numerical gradient estimation." Physical Review Letters 95, no. 5 (2005): 050501.

[2] Tongyang Li, Shouvanik Chakrabarti, and Xiaodi Wu. "Sublinear quantum algorithms for training linear and kernel-based classifiers." In International Conference on Machine Learning, pp. 3815-3824. PMLR, 2019. arXiv:1904.02276

[3] Srinivasan Arunachalam and Reevu Maity. "Quantum boosting." In International Conference on Machine Learning, pp. 377-387. PMLR, 2020. arXiv:2002.05056

[4] Tongyang Li, Chunhao Wang, Shouvanik Chakrabarti, and Xiaodi Wu. "Sublinear classical and quantum algorithms for general matrix games." In Proceedings of the AAAI Conference on Artificial Intelligence, vol. 35, no. 10, pp. 8465-8473. 2021.

[5] Adam Bouland, Yosheb M. Getachew, Yujia Jin, Aaron Sidford, and Kevin Tian. "Quantum speedups for zero-sum games via improved dynamic Gibbs sampling." In International Conference on Machine Learning, pp. 2932-2952. PMLR, 2023. arXiv: 2301.03763

[6] Minbo Gao, Zhengfeng Ji, Tongyang Li, and Qisheng Wang. "Logarithmic-Regret Quantum Learning Algorithms for Zero-Sum Games." To appear in the Conference on Neural Information Processing Systems, 2023. arXiv:2304.14197

---

> ### Author Response · Authors · 2023-11-18
>
> Regarding the novelty of our paper, for the algorithmic part, the hybrid algorithm we propose in the paper achieves a quadratic speedup on a novel and important problem in optimization. Notably, to the best of our knowledge, our algorithm is the first one that applies the quantum Gibbs sampling technique into gradient-based classical algorithms. Moreover, the implementation of our algorithm relies on the crucial observation that a key part of the original classical algorithm, namely the sampling of index of coordinates, is the bottleneck of the algorithm and can be sped up by using a quantum preparing algorithm.
> We believe that this improvement is non-trivial since merely preparing separately several states is not enough for providing the quadratic speedup.
> To improve the algorithm from $O(N)$ to $O(\sqrt{N})$, we would need to construct a circuit as shown in Figure 1 in our paper, which is used later as a subroutine of the batch state preparation in line 1 of Algorithm 1.
>
> Moreover, the proof of our lower bound is non-trivial and is technically very different from existing quantum lower bounds for optimization problems, see e.g., [7, 8].
> Basically, our lower bound considers the setting where a realistic loss function $\max_i f_i$ is to be minimized by an algorithm provided with oracles $O_i$ for each function $f_i$.
> To the best of our knowledge, we are the first to consider such a setting.
> As a result, the "chain" structure we used to prove the lower bound is more delicate than other chains used in, e.g., the proof of [9].
> We also introduced a "multi-round" unstructured search problem and combine it with the chain structure of the hard instance.
> As far as we know, this technique is novel and also allows for an elegant proof for the lower bound of this problem.
> This makes our proof non-trivial since if one is to approach this problem with known techniques, the lower bound would not achieve the tightness our result with respect to $N$.
>
> **Reference**
>
> [7] Ankit Garg, Robin Kothari, Praneeth Netrapalli, and Suhail Sherif. "No Quantum Speedup over Gradient Descent for Non-Smooth Convex Optimization." In 12th Innovations in Theoretical Computer Science Conference (ITCS 2021). arXiv:2010.01801
>
> [8] Chenyi Zhang and Tongyang Li. "Quantum Lower Bounds for Finding Stationary Points of Nonconvex Functions." In International Conference on Machine Learning, pp. 41268-41299. PMLR, 2023. arXiv:2212.03906
>
> [9] Yair Carmon, Arun Jambulapati, Qijia Jiang, Yujia Jin, Yin Tat Lee, Aaron Sidford, and Kevin Tian. "Acceleration with a ball optimization oracle." Advances in Neural Information Processing Systems 33 (2020): 19052-19063.

---

### Meta-Review · Area_Chair_w94z · 2023-12-07

**Metareview:**

The paper considers the problem of minimizing the maximum of $N$ convex Lipschitz function in the quantum oracle model of computation. The starting point for the work are the recent results by Carmon et al. using ball oracle acceleration to solve this problem to error $\epsilon > 0$ with oracle complexity $O(N/\epsilon^{2/3} + 1/\epsilon^{8/3})$ in the *classical* oracle model (the same paper provided an $\Omega(N/\epsilon^{2/3})$ oracle complexity lower bound).

The present paper makes a neat observation, which relates the regularized ball optimization oracle to the Gibbs sampling problem. Based on this observation, the paper leverages quantum Gibbs sampling to obtain the quantum speedup. The resulting quantum oracle complexity is $O(\sqrt{N}/\epsilon^{5/3} + 1/\epsilon^{8/3}),$ which is (quadratically) better in terms of the dependence on $N$ but worse in terms of the dependence on $1/\epsilon$ (this is not unusual for quantum speed ups in the oracle model of optimization). The paper also provides a quantum oracle lower bound $\Omega(\sqrt{N}/\epsilon^{2/3})$ and leaves open the possibility of reducing the dependence on $1/\epsilon$ in the upper bound.

Overall, the paper is presented well, the ideas are clear, and it is a solid result. Thus, I recommend acceptance.

**Justification For Why Not Higher Score:**

I wouldn't mind a higher score, but I did not see enough support from the reviewers to recommend it.

**Justification For Why Not Lower Score:**

It seems like a clear accept, based on the results and the reviews.

---

### Decision · Program_Chairs · 2024-01-16

Accept (poster)